# Class Distribution Shifts in Zero-Shot Learning: Learning Robust Representations

**Yuli Slavutsky**
Department of Statistics and Data Science
The Hebrew University of Jerusalem
Jerusalem, Israel
yuli.slavutsky@mail.huji.ac.il

**Yuval Benjamini**
Department of Statistics and Data Science
The Hebrew University of Jerusalem
Jerusalem, Israel
yuval.benjamini@mail.huji.ac.il

## Abstract

Zero-shot learning methods typically assume that the new, unseen classes encountered during deployment come from the same distribution as the the classes in the training set. However, real-world scenarios often involve class distribution shifts (e.g., in age or gender for person identification), posing challenges for zero-shot classifiers that rely on learned representations from training classes. In this work, we propose and analyze a model that assumes that the attribute responsible for the shift is unknown in advance. We show that in this setting, standard training may lead to non-robust representations. To mitigate this, we develop an algorithm for learning robust representations in which (a) synthetic data environments are constructed via hierarchical sampling, and (b) environment balancing penalization, inspired by out-of-distribution problems, is applied. We show that our algorithm improves generalization to diverse class distributions in both simulations and experiments on real-world datasets.

## 1   Introduction

Zero-shot learning systems [14, 27] are designed to classify instances of new, previously unseen classes at deployment, a scenario known as *open-world* classification. These systems are widely applied in extreme multi-class applications, such as face or voice recognition [19] for matching observations of the same individual, and more generally, for learning data representations [2].

Class distribution shifts typically refer to changes in the prevalence of a fixed set of classes between training and testing. In zero-shot learning, however, a different challenge arises: the appearance of entirely new classes at test time. This raises a critical question – are these new classes drawn from the same distribution as the training classes? Most zero-shot methods assume that they are, an assumption that not only shapes the design of test sets [57, 16] but also plays an explicit role in assessing the generalization capabilities of zero-shot classifiers [59, 48].

In practice, training classes are often chosen based on convenience and accessibility during data collection. Even when data is carefully collected, the distribution of classes may shift over time, leading to a different distribution. For instance, this could occur when a face recognition system is deployed in a building located in a neighborhood undergoing demographic changes.

Class distribution shifts pose significant challenges to zero-shot classifiers, since they rely on learning data representations from the training classes to distinguish new, unseen ones. Typically, these classifiers are trained by minimizing the loss on the training set to effectively separate the training classes. However, this approach may result in poor performance when confronted with data from distributions that differ significantly from the class distribution in the training data. Notably, in person re-identification, this concern gained attention from a fairness perspective with respect to gender

38th Conference on Neural Information Processing Systems (NeurIPS 2024).

[15, 23], age [5, 33, 50], and racial [39, 54] bias. In all these studies the variable (i.e., gender, age, race) expected to cause the distribution shift was known in advance.

In contrast, in real-world scenarios, the attribute responsible for a future distribution shift is usually unknown during training. In such cases, existing approaches based on collecting balanced datasets or re-weighting training examples [54, 41, 53] are inapplicable. Furthermore, while class distribution shifts have been extensively studied in the standard setting of supervised learning (see Appendix A), previous research assumed a *closed-world* setting that does not account for new classes at test time. Instead, it only addressed changes in the prevalence of fixed classes between training and testing. Consequently, class distribution shifts in zero-shot learning remain largely unaddressed.

In this paper we first address these limitations by examining the effects of class distribution shifts on constrastive zero-shot learning, by proposing and analyzing a parametric model (§3). We identify conditions where minimizing loss in this model leads to representations that perform poorly when a distribution shift has occurred.

We then use the insights gained from this model to present our second contribution (§4): an algorithm for learning representations that are robust against class distribution shifts in zero-shot classification. In our proposed approach, *artificial data environments* with diverse attribute distributions are constructed using hierarchical subsampling, and an *environment balancing* criterion inspired by out-of-distribution (OOD) methods is applied. We assess our method's effectiveness in both simulations and experiments on real-world datasets, demonstrating its enhanced robustness in §5.

## 1.1 Problem Setup

Let $\{z_i, c_i\}_{i=1}^{N_z}$ be a labeled set of training data points $z \in \mathcal{Z}$ and classes $c \in \mathcal{C}$, such that $c_i$ is the class of $z_i$.

In this work, we focus on verification algorithms that enable *open-world* classification by determining whether two data points $x_{ij} := (z_i, z_j)$ belong to the same class. For instance, in person re-identification the task is to identify whether two data points (e.g., face images or voice recordings) belong to the same person. We denote this by $y_{ij}$, where $y_{ij} = 1$ if $c_i = c_j$ and $y_{ij} = 0$ otherwise. When the identity of each data point in the pair is not important, a single index is used for simplicity, namely $(x_k, y_k)$.

We assume that each class $c$ is characterized by some attribute $A$. We further assume that the training classes are sampled from $P_C(c)$, the test classes are sampled according to $Q_C(c)$, and the two distributions differ solely due to a shift in the distribution of an attribute $A$:

$$P_C(c) = \int P_{C|A}(c|a)\boldsymbol{P}_A(a)\, da, \quad Q_C(c) = \int P_{C|A}(c|a)\boldsymbol{Q}_A(a)\, da. \tag{1}$$

Importantly, we assume that the attribute $A$ is unknown, and that both during training and testing, data points $z \in \mathcal{Z}$ for each class are sampled according to $P_{Z|C}(z|c)$. For instance, revisit the person identification example where each person is a class. If the attribute $A$ is binary (e.g., $a_1$ is blond and $a_2$ is dark-haired), then $P(C|A = a_1)$ represents the distribution of people with blond hair, and $P(C|A = a_2)$ of individuals with other hair colors. The training classes might be predominantly sampled from the blond population $P(A = a_1) = \rho_{\text{tr}} = 0.8$, while test classes are predominantly sampled from $Q(A = a_1) = \rho_{\text{te}} = 0.1$.

We focus on verification techniques based on *deep metric learning* methods (for surveys see [43, 34]) such as contrastive-learning [17], Siamese neural networks [24], triplet networks [20], and other more recent variations [35, 49, 56, 58]. These methods learn a representation function that maps data points to a representation space $g : \mathcal{Z} \to \hat{\mathcal{Z}}$, so that examples from the same class are close (in a predefined distance function $d(\cdot, \cdot)$), while those from different classes are farther apart.

We assume that $g$ is a neural network trained by optimizing a deep-metric-learning loss, such as the contrastive loss [17]:

$$\ell(z_i, z_j, y_{ij}; d_g) := y_{ij} d_g^2(z_i, z_j) + (1 - y_{ij}) \max\{0, m - d_g(z_i, z_j)\}^2 \tag{2}$$

where $m \geq 0$ is a predefined margin, and $d_g(z_i, z_j) := d(g(z_i), g(z_j))$ is the distance between the representations of the datapoints $z_i, z_j$. In our theoretical analysis, we examine the no-hinge

contrastive loss (see Appendix B for additional details):

$$\widetilde{\ell}(z_i, z_j, y_{ij}; d_g) := y_{ij} d_g^2(z_i, z_j) + (1 - y_{ij})(m - d_g(z_i, z_j))^2. \tag{3}$$

To evaluate the class separation capability of a representation, we treat the distances between representations, $d_g(z_i, z_j)$, as classification scores. Following common practice in the field (e.g., [47, 22]), we use the area under the receiver operating characteristic curve (AUC) to evaluate the representation, enabling threshold-agnostic assessment:

$$AUC(g) := P(d_g(z_i, z_j) < d_g(z_u, z_v) | y_{ij} = 1, y_{uv} = 0). \tag{4}$$

Our goal is to learn a representation $g$ that is robust to class attribute shifts. That is, such that for an unknown shifted distribution $Q_A$, the performance $\mathbb{E}_{Q_A}[\text{AUC}(g)]$ does not significantly deteriorate compared to the performance obtained on the training distribution $P_A$.

## 2   Background on Environment Balancing Methods in OOD Generalization

The field of OOD generalization gained attention since the work of Peters et al. [36], [37], which deals with closed-world classification where training data is gathered from multiple environments $E_{\text{train}}$. In this setting it is assumed that in each environment $e \in E_{\text{train}}$ examples share the same joint distribution $P_{C,Z}^e(c, z)$, but across environments the joint distribution changes, often due to variations in $P_{Z|C}^e(z|c)$. A well-known example [1] involving the classification of images of cows and camels demonstrates how an algorithm relying on background cues during training (e.g., cows in green pastures, camels in deserts) performs poorly on new images of cows with sandy backgrounds.

Several approaches that rely on access to diverse training environments were proposed to identify stable relations between the data point $z$ and its class $c$. Examples of such stable relations include choosing causal variables using statistical tests [42], leveraging conditional independence induced by the common causal mechanism [9], and using multi-environment calibration as a surrogate for OOD performance [52].

Most relevant to our work are methods that aim to balance the loss over multiple environments. These methods consider a representation $g = g_\theta$ that is a neural network parameterized by $\theta$ trained to optimize an objective of the form

$$\min_\theta \sum_{e \in E_{\text{train}}} \ell^e(g_\theta) + \lambda R(g_\theta, E_{\text{train}}) \tag{5}$$

where $\ell^e(g_\theta)$ is the empirical loss obtained on the environment $e$, $E_{\text{train}}$ is the set of all training environments, $R$ is a regularization term designed to balance performance over multiple environments, and $\lambda$ is a regularization factor balancing the tradeoff between the empirical risk minimization (ERM) term and the balance penalty. Below, we describe three such methods, which we will refer to later in the paper.

**Invariant risk minimization (IRM)** presented in [1], aims to find data representations $g_\theta$ such that the optimal classifier $w$ on top of the data representation $w \circ g_\theta$ is shared across all environments. Therefore, the authors proposed minimizing the sum of environment losses $\ell^e(w \circ g_\theta)$ over all training environments such that $w \in \arg\min_{w'} \ell^e(w' \circ g_\theta)$ for all $e \in E_{\text{train}}$. However, since this objective is too difficult to optimize, a relaxed version was also proposed, taking the form of Equation 5 with a penalty that measures how close $w$ is to minimizing $\ell^e(w \circ g_\theta)$: $R_{\text{IRMv1}}^e(g_\theta) = \left\| \nabla_{w|w=1} \ell^e(w \cdot g_\theta) \right\|^2$.

Note that for loss functions for which optimal classifiers can be expressed as conditional expectations, the original IRM objective is equivalent to the requirement that for all environments $e, e' \in E_{\text{train}}$, $\mathbb{E}_{P_{C,Z}^e}[c|g(z) = h] = \mathbb{E}_{P_{C,Z}^{e'}}[c|g(z) = h]$, where $P_{C,Z}^e$ and $P_{C,Z}^{e'}$ are the joint data distributions in the respective environments.

**Calibration Loss Over Environments (CLOvE)** presented in [52], leverages the equivalence above to establish a link between multi-environment calibration and invariance for binary predictors ($c \in \{0, 1\}$). The proposed regularizer is based on the *maximum mean calibration error* (MMCE) [26]. Let $s : \hat{\mathcal{Z}} \to [0, 1]$ be a classification score function applied on the representation $s \circ g$, and

$s_i = \max\{s \circ g(z_i), 1 - s \circ g(z_i)\}$ be the *confidence* on the $i$-th data point. Denote the *correctness* as $b_i = \mathbb{1}\{|c_i - s_i| < \frac{1}{2}\}$, and let $K : \mathbb{R} \times \mathbb{R} \to \mathbb{R}$ be a universal kernel. Let $Z^e$ denote the training data in the environment $e$. The authors proposed using the MMCE as a penalty in an objective that takes the form of Equation 5 with $R_{\text{MMCE}}^e(s, g_\theta) = \frac{1}{m^2} \sum_{z_i, z_j \in Z^e} (b_i - s_i)(b_j - s_j) K(s_i, s_j)$.

**Variance Risk Extrapolation (VarREx)** proposed by Krueger et al. [25], is based on the observation that reducing differences in loss (risk) across training domains can reduce a model's sensitivity to a wide range of distribution shifts. The authors found that using the variance of losses as a regularizer is stabler and more effective compared to other penalties. Therefore, they propose the following regularization term for $n$ training environments: $R_{\text{VarREx}}(g_\theta, E_{\text{train}}) = \text{Var}\left(\ell^{e_1}(g_\theta), \ldots, \ell^{e_n}(g_\theta)\right)$.

While simple and intuitive, this approach assumes that losses across different environments accurately reflect the classifier's performance. However, as discussed in §4, this is often not true for deep metric learning losses, where significant changes in loss may correspond to only minor variations in performance.

# 3 Parametric Model of Class Distribution Shifts in Zero-Shot Learning

In this section, we introduce a parametric model of class distribution shifts. Our model shows that in zero-shot learning, even if the conditional distribution of data given the class $P(z|c)$ remains the same between training and testing, a shift in the class distribution from $P(c)$ to $Q(c)$ can cause poor performance on newly encountered classes sampled from the shifted distribution $Q(c)$.

Assume that for all classes, the data points $z_i \in \mathbb{R}^d$ are sampled from $z_i|c_i \sim \mathcal{N}(c_i, \Sigma_z)$, where $\Sigma_z = \nu_z \cdot I_d$, and $I_d$ is the identity matrix. Let the attribute $A$ indicate the type of a class $c$, with two possible types: $a_1$ and $a_2$. Assume that the classes $c_i$ are drawn according to $c_i \sim \mathcal{N}(0, \Sigma_a)$ for $a \in \{a_1, a_2\}$. Finally, assume that in training, $a_1$ is the majority type with $P(a_1) = \rho_{\text{tr}} \gg 0.5$, whereas at test time, $a_2$ is the majority type with $Q(a_1) = \rho_{\text{te}} \ll 0.5$.

We construct the model such that differences between the training class distribution $P(c)$ and the test distribution $Q(c)$ stem solely from a shift in the mixing probabilities of an unknown attribute $A$ (see Equation 1). Therefore, we define $\Sigma_a$ as a diagonal matrix with replicates of three distinct values on its diagonal: $\nu_0, \nu^+, \nu^-$. Let $0 < \nu^- < \nu_z \leq \nu_0 < \nu^+$. Then, in the coordinates corresponding to $\nu_0$ and $\nu^+$ data points from different classes are well separated, whereas in the coordinates corresponding to $\nu^-$ they are not. Assume the coordinates corresponding to $\nu_0$ are shared by both types, but $\nu^+$ and $\nu^-$ are swapped:

$$\Sigma_{a_1} = \text{diag}\left(\overbrace{\nu_0, \ldots, \nu_0}^{d_0}, \overbrace{\nu^+, \ldots, \nu^+}^{d_1}, \overbrace{\nu^-, \ldots, \nu^-}^{d_2}\right),$$
$$\Sigma_{a_2} = \text{diag}\left(\nu_0, \ldots, \nu_0, \nu^-, \ldots, \nu^-, \nu^+, \ldots, \nu^+\right).$$

An illustration with one replicate of each value is shown in Figure 1.

The following proposition shows that if the number of dimensions $d_1$ that allow good separation for classes of type $a_1$ is relatively similar to the number of dimensions $d_2$ that enable good separation for classes of type $a_2$, specifically if $h_l(\rho, \nu_z, \nu_0, \nu_1, \nu_2) < \frac{d_2 + 2}{d_1 + 2} < h_u(\rho, \nu_z, \nu_0, \nu_1, \nu_2)$, then the optimal solution for the training distribution prioritizes the components (features) corresponding to $\nu^+$ for classes of type $a_1$. Thus, the prioritized features allow good separation for classes from the majority type in training, but offer poor separation for the shifted test distribution, where most classes are of type $a_2$. Note that if $d_2$ is large, when combined, the corresponding components may still provide reasonable separation. We define $h_l$ and $h_u$ in Equation 35 and provide the proof of Proposition 1 in Appendix B.2.

**Proposition 1.** *Consider a weight representation $g(z) = Wz$, where $W \in \mathbb{R}^{d \times d}$ is a diagonal matrix, and the squared Euclidean distance $d_g(z_i, z_j) = \|W(z_i - z_j)\|^2$. Let $W^* = \text{diag}(w^*) \in \arg\min_W \mathbb{E}\left[\widetilde{\ell}(\cdot, \cdot, \cdot; d_g)\right]$. Denote $\overline{w}_1^{*2} = \frac{1}{d_1} \sum_{k=d_0+1}^{d_1} w_k^{*2}$ and $\overline{w}_2^{*2} = \frac{1}{d_2} \sum_{k=d_1+1}^{d} w_k^{*2}$. Then, for all $\rho > \frac{1}{2}$ and $\nu_z, \nu_0, \nu_1, \nu_2, d_1, d_2$ satisfying $h_l(\rho, \nu_z, \nu_0, \nu_1, \nu_2) < \frac{d_2+2}{d_1+2} < h_u(\rho, \nu_z, \nu_0, \nu_1, \nu_2)$ it holds that $d_2 \overline{w}_2^{*2} \leq d_1 \overline{w}_1^{*2}$.*

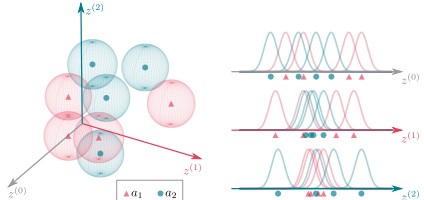

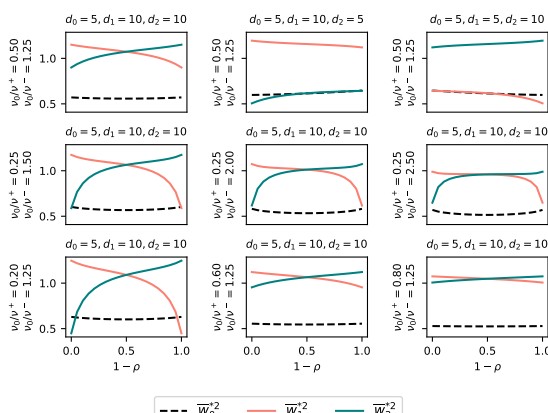

Figure 1: Illustration of the parametric model. Classes of each type are best separated along specific axes: classes of type $a_1$ along the red axis ($z^{(1)}$) and classes of type $a_2$ along the green axis ($z^{(2)}$). On axis $z^{(0)}$ both types can be separated but not as effectively as on their respective optimal axes.

Figure 2: Optimal weights. Top row: $d_0$ is fixed, $d_1$ and $d_2$ vary. Middle and bottom rows: $d_0, d_1, d_2$ are fixed. Middle: $\nu_0/\nu^-$ varies. Bottom: $\nu_0/\nu^+$ varies.

Note that the conditions outlined in Proposition 1 are sufficient but not necessary. Accordingly, in Appendix B.3, we provide the complete analytical solution for $w^*$ that minimizes the expected loss $\mathbb{E}\left[\widetilde{\ell}\left(\cdot, \cdot, \cdot; d_g\right)\right]$ for the weight representation $g(z) = Wz$, using the squared Euclidean distance. According to Proposition 1, larger $d_2$ values favor $\nu^-$ for better aggregated separation. Increasing $\nu_0/\nu^+$ leads to increased differences between $w_1^{*2}$ and $w_2^{*2}$, and vice versa for $\nu_0/\nu^-$.

These relationships in the optimal solution are illustrated in Figure 2, showcasing different scenarios. The top row shows that when $d_1 = d_2 = 10$ dimensions favoring classes of type $a_1$ are prioritized for $\rho > 0.5$, while those favoring type $a_2$ are prioritized for $\rho < 0.5$. When $d_1 = 10$ while $d_2 = 5$, dimensions favoring type $a_1$ are prioritized for all values of $\rho$, and vice versa when $d_2$ is significantly larger than $d_1$. The middle and the bottom row further explore the $d_1 = d_2$ case, showing how differences in separability between shared dimensions ($\nu_0$) and type-favoring dimensions impact weight allocation.

Since components corresponding to $\nu^+$ for classes of type $a_1$ align with $\nu^-$ for classes of type $a_2$, the optimal representation for the training distribution results in poor separation for the shifted test distribution. Therefore, a robust representation should prioritize dimensions that provide effective separation for both class types, corresponding to $\nu_0$.

This aligns with a common principle in the OOD generalization field, where robust representations are those that rely on features shared across environments (see §2). This principle is often referred to as *invariance*.

## 4   Proposed Approach

Motivated by our analysis of the parametric model, we propose a new approach for tackling class distribution shifts in zero-shot learning. Our approach revolves around two key ideas: (i) during training, different mixtures of the attribute $A$ can be produced by sampling small subsets of the classes, forming artificial environments, and (ii) penalizing for differences in performance across these environments is likely to increase robustness to the class mixture encountered at test time.

### 4.1   Synthetic Environments

Standard ERM training involves sampling pairs of data points $(z_i, z_j)$ uniformly at random from all $N_c$ classes available during training. However, as discussed in §3, this is prone to overfitting to the attribute distribution of the training data. Since the identity of the attribute is unknown, weighted sampling (and similar approaches) cannot be used to create environments with different attribute mixtures.

Yet, our goal is to design artificial environments with diverse compositions of the (unknown) attribute of interest. To do so, we leverage the variability in small samples: while class subsets of similar size to $N_c$ maintain attribute mixtures similar to the overall training set, smaller subsets with $k \ll N_c$ classes are likely to exhibit distinct attribute mixtures. Therefore, we propose creating multiple environments, composed of examples from few sampled classes.

This results in a hierarchical sampling scheme for the data pairs: first, sample a subset of $k$ classes, $S = \{c_1, \ldots, c_k\}$. Then, for each $c \in S$ sample $2r$ pairs of data points as follows: $r$ pairs from within the class $c$, $\{z_i; c_i = c\}$, uniformly at random (positive pairs); and $r$ negative pairs, where one point is sampled uniformly at random from $c$, and the other from all other data points in $S$, $\{z_i; c_i \neq c, c_i \in S\}$.[1]

Across multiple class subsets $S = \{S_1, \ldots, S_n\}$, this hierarchical sampling results in diverse mixtures of any unknown attribute (see Figure 3). In particular, in some of the class subsets, classes from the overall minority type constitute the majority in the environment.

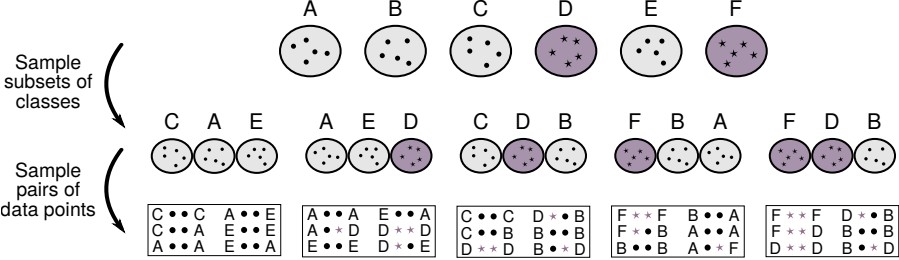

Figure 3: Illustration of the proposed hierarchical sampling. Top: $N_c = 6$ classes, with 2 minority-type classes D, F (in purple). Middle: synthetic environments formed by sampling small ($k = 3$) class subsets; in $1/5$ of the environments, minority-type classes become the majority constituting $2/3$ of the classes. Bottom: sampling $r = 1$ positive and $r = 1$ negative pairs for each class in the environment.

## 4.2 Environment Balancing Algorithm for Class Distribution Shifts

Our goal is to learn data representations that will allow separation between classes without knowing which attribute is expected to change and how significantly. Therefore, we require the learned data representation to perform similarly well on all mixtures obtained on the synthetic environments.

To achieve this, inspired by OOD performance balancing methods (see §2), we optimize a penalized objective:

$$\min_\theta \quad \sum_{l=1}^{n} \ell^{S_l}(g_\theta) + \lambda R(S_1, \ldots, S_n) \tag{6}$$

where $R(S_1, \ldots, S_n)$ is any balancing term between the constructed synthetic environments.

Note that computing $R(S_1, \ldots, S_n)$ often involves evaluating some value on each environment separately. For a general balancing term, we denote the value in the $l$-th environment as $f(S_l)$ and accordingly express $R(S_1, \ldots, S_n) = \mathring{f}(f(S_1), \ldots, f(S_n))$, where $\mathring{f}$ represents the corresponding aggregation function. Our approach[2] for balancing performance across synthetic environments of class subsets, is outlined in Algorithm 1.

## 4.3 Balancing Performance Instead of Loss

In multiple OOD penalties (e.g., IRM and VarREx), $f$ represents the loss in each environment, which, in deep metric learning algorithms, is based on distance. This presents a challenge in zero-

---

[1] Here, for simplicity we create balanced environments, but different proportions of positive examples can be considered instead.

[2] For notation simplicity we assume that the unpenalized training loss is applied to pairs of data points $(x_{ij}, y_{ij}) = ((z_i, z_j), \mathbb{1}_{c_i = c_j})$, but it can easily be adapted for any tuple size (e.g., triplets).

---

**Algorithm 1** Robust Zero-Shot Representation

---

**Input:** Labeled data $D = \{z_i, c_i\}_{i=1}^{N_z}$, number of synthetic environments $n$, number of classes within subset $k$, number of pairs per class $2r$, neural network $g(\cdot; \theta)$, loss $\ell$, distance function $d$, regularization functions $f$, $\mathring{f}$, initial weights $\theta_0$, number of training iterations $T$, learning rate $\eta$

**Output:** Learned representation $g(\cdot; \theta_T)$

Compute unique classes $C^* = \{c^{(1)}, \ldots, c^{(N_c)}\}$

**for** $t = 1$ **to** $T$ **do**

    **for** $l = 1$ **to** $n$ **do**

        Sample $k$ classes from $C^*$ without replacement: $S_l^{(t)} = \{c_l^{(1)}, \ldots, c_l^{(k)}\}$.

        From each class in $S_l^{(t)}$ sample $r$ positive and $r$ negative data pairs. Denote the set by $D_l^{(t)}$.

        Compute $f(S_l^{(t)})$.

        Compute average unpenalized loss over $(x_m, y_m) \in D_l^{(t)}$:     $\bar{\ell}_l^{(t)} = \frac{1}{2rk} \sum_{m=1}^{2rk} \ell(x_m, y_m)$.

    **end for**

    Compute $R^{(t)} := R\left(S_1^{(t)}, \ldots, S_n^{(t)}\right) = \mathring{f}\left(f(S_1^{(t)}), \ldots, f(S_n^{(t)})\right)$.

    Update network parameters performing a gradient descent step:

    $\theta^{(t)} \leftarrow \theta^{(t-1)} - \eta \nabla_\theta \left(\frac{1}{n} \sum_{l=1}^{n} \bar{\ell}_l^{(t)} + R^{(t)}\right)$

**end for**

**Return:** $g(\cdot; \theta_T)$

---

shot verification, where sampled tuples often include numerous easy negative examples, leading to performance plateau early in the learning process, although the distances themselves still exhibit considerable variations. Strategies like selecting the most difficult tuples [18] were proposed to address this issue, however these methods have been found to generate noisy gradients and loss values [34].

We therefore propose to balance performance directly instead of relying on the losses in the training environments. Denote the set of negative pairs in a synthetic environment by $D_l^0 = \{x_{ij} = (z_i, z_j) : c_i, c_j \in S_l, y_{ij} = 0\}$ and the set of positive pairs by $D_l^1 = \{x_{ij} = (z_i, z_j) : c_i, c_j \in S_l, y_{ij} = 1\}$. An unbiased estimator of the AUC on a given synthetic environment $S_l$ is given by

$$\widetilde{\text{AUC}}(S_l; d_g) = \frac{1}{|D_l^0| \, |D_l^1|} \sum_{x_{ij}} \sum_{x_{uv}} \mathbb{1}\left[d_g(x_{ij}) < d_g(x_{uv})\right] \tag{7}$$

for $x_{ij} \in D_l^1$ and $x_{uv} \in D_l^0$. Since this estimator is non-differentiable and therefore cannot be used in gradient-descent-based optimization, we use *soft-AUC* as an approximation [7]

$$\widehat{\text{AUC}}(S_l; d_g) \frac{1}{|D_l^0| \, |D_l^1|} \sum_{x_{ij}} \sum_{x_{uv}} \sigma_\beta\left(d_g(x_{uv}) - d_g(x_{ij})\right) \tag{8}$$

where a sigmoid $\sigma_\beta(t) = \frac{1}{1 + e^{-\beta t}}$ approximates the step function. Note that when $\beta \to \infty$, $\sigma_\beta$ converges pointwise to the step function. Consequently, we propose the penalty:

$$R_{\text{VarAUC}}(S_1, \ldots, S_n; g_d) = \widehat{\text{Var}}\left(\widehat{\text{AUC}}(S_1; g, d), \ldots, \widehat{\text{AUC}}(S_n; g, d)\right). \tag{9}$$

### 4.4 How Many Environments Are Needed?

The proposed hierarchical sampling scheme allows for the construction of many synthetic environments with various attribute mixtures, influenced by the number of classes in each environment. As shown in the analysis below, this ensures that with high probability there will be at least one environment with a pair of minority type classes, thereby supporting learning to separate negative pairs within the minority type.

In each training iteration, we consider $n$ class subsets (environments) of size $k$. Our goal is to achieve robustness to all attribute values $a$ that are associated with at least $\rho_{\min} \in (0, 1)$ of the training classes. Note that $\rho_{\min}$ is specified by the practitioner without knowledge of the true attribute that may cause the shift or its true prevalence $\rho$ in the training set.

We compute the number of synthetic environments $n$, such that with high probability of $(1 - \alpha)$, $S_1, \ldots, S_n$ will include at least one subset with at least two classes associated with $a$ (otherwise none of the subsets would contain negative pairs with the attribute $a$). Denote the probability of a given subset not to contain any class associated with $a$ by $\phi_0 = \binom{\lceil (1-\rho_{\min})N_c \rceil}{k} / \binom{N_c}{k}$ and the probability of a given subset to contain exactly one such class by $\phi_1 = \rho_{\min}N_c \binom{\lceil (1-\rho_{\min})N_c \rceil}{k-1} / \binom{N_c}{k}$. Therefore, the required number of environments needed to ensure that at least two minority-type classes appear together in the same environment is

$$n \approx \frac{\log(\alpha)}{\log(\phi_0 + \phi_1)}. \tag{10}$$

Note that that $n$ is typically much smaller than $\binom{N_c}{k}$.

## 5  Empirical Results

Our method enhances standard training with two components: a hierarchical sampling scheme and a balancing term for synthetic environments. To the best of our knowledge, this is the first work addressing OOD generalization for class distribution in zero-shot learning. We therefore benchmark our algorithm against the ERM baseline (uniform random sampling with an unpenalized score) and a hierarchical sampling baseline (hierarchical sampling with unpenalized score).

Additionally, we tested standard regularization techniques including dropout and the $L_2$ norm, which did not yield notable improvements in the distribution shift scenario, and are therefore not shown.

To ensure a comprehensive comparison, in addition to the proposed VarAUC penalty, we evaluate variants of our algorithm in which the IRM, CLOvE, and VarREx penalties are used instead. While we show that VarAUC consistently outperforms other penalties, the crucial improvement lies in its performance compared to the ERM baseline: application of existing OOD penalties is enabled by the construction of synthetic environments in our algorithm. As discussed in Appendix C, this construction facilitates the formulation of class distribution shifts in zero-shot learning within the OOD setting.

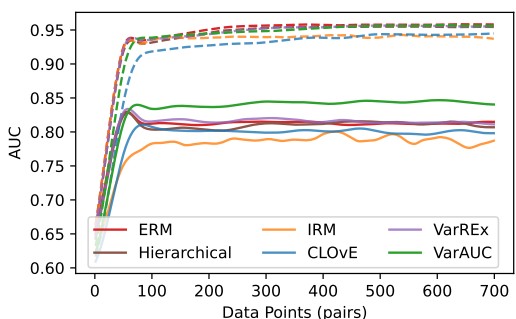

Figure 4: Average AUC over 10 simulation repetitions for majority attribute proportion $\rho = 0.9$ in training (and 0.1 in test). Solid lines: distribution-shift. Dashed lines: in-distribution. Our method improves robustness for shifts, without compromising training distribution results.

In all of the experiments performed, we trained the network with contrastive loss (Equation 2) and the normalized cosine distance: $d_g(z_1, z_2) = \frac{1}{2}\left(1 - \frac{g(z_1)\cdot g(z_2)}{\|g(z_1)\|\|g(z_2)\|}\right)$. The specific setups are detailed below (additional details can be found in Appendix F), and code to reproduce our results is available at https://github.com/YuliSl/Zero_Shot_Robust_Representations .

### 5.1  Simulations: Revisiting the Parametric Model

We now revisit the parametric model presented in §3. To increase the complexity of the problem, we add dimensions where classes from both types are not well separated. That is, $\Sigma_a$ includes additional dimensions set to zero.

**Setup** We used 68 subsets in each training iteration, each consisting of two classes. This corresponds to choosing $\rho_{\min} = 0.1$ (desired sensitivity, regardless of the true unknown parameter $\rho \in \{0.05, 0.1, 0.3\}$), with a low $\alpha$ value of 0.5, resulting in the construction of fewer environments according to Equation 10. For each class, we sampled $2r = 10$ pairs of data points. The representation was defined as $g(z) = wz$ for $w \in \mathbb{R}^{d \times p}$ [3]. Here we focus on the case of $p = 16$, $\nu_z = \nu_0 = 1$,

---

[3]A linear representation is chosen to facilitate an analysis of the learned representation space.

$\nu^- = 0.1$, $\nu_+ = 2$, $d_0 = 5$, $d_1 = d_2 = 10$. The results for additional representation sizes $p$, noise ratios $\frac{\nu^+}{\nu^-}$ and varying proportions of positive and negative examples are presented in Appendix D.1.

To assess the importance assigned to each dimension, we examine weight values relative to other weights: $\text{Importance}_i = \left| \frac{\sum_{j=1}^{p} w_{ij}}{\sum_{i'=1}^{d} \sum_{j'=1}^{p} w_{i'j'}} \right|$. (11)

**Results** In Figure 5 we examine the learned representation. The analysis indicates that ERM prioritizes dimensions 5-15, providing good separation for $a_1$, the dominant type in training,

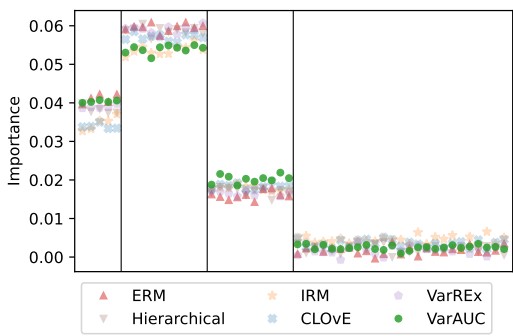

but leading to poor separation after the shift. ERM assigns low weights to dimensions beneficial for both types (0-5) and those suitable for $a_2$ (15-25). In contrast, our algorithm, particularly with the two variance-based penalties, assigns the lowest weights to dimensions corresponding to $a_1$ and higher weights to shared dimensions and those that effectively separate $a_2$ classes.

In Figure 4, the learning progress is depicted for $\rho = 0.9$ (a similar analysis for $\rho = 0.95$ and $\rho = 0.7$ can be found in Appendix D). Performance on the same distribution as the training data is similar for ERM and our algorithm, suggesting that applying our algorithm does not negatively impact performance when no distribution shift occurs. However, when there is a distribution shift our algorithm achieves much better results. The VarREx penalty achieves high AUC values more quickly than the VarAUC penalty, but the

Figure 5: Average feature importance for $\rho = 0.9$, 10 repetitions. Our VarAUC penalty favors shared features (blocks 1 and 3), while deprioritizing majority features (block 2). All methods assign low weight to noise features (block 4).

VarAUC penalty attains higher overall accuracy. IRM shows noisier convergence, since it is applied directly on the gradients, which have been shown to be noisy in contrastive learning due to high variance in data-pair samples [34]. Means and standard deviations are reported in Appendix D.1, as well as the results for additional data dimensions, positive proportions, and variance ratios.

## 5.2 Experiments on Real Data

**Experiment 1 - Species Recognition** We used the ETHEC dataset [11] which contains 47,978 butterfly images from six families and 561 species (example of the images are provided in Appendix D). We filtered out species with less than five images and focused on images of butterflies from the Lycaenidae and Nymphalidae families. In the training set, $10\%$ of the species were from the Nymphalidae family, while at test time, $90\%$ of the species were from the Nymphalidae family. For each class we sampled $2r = 20$ pairs.

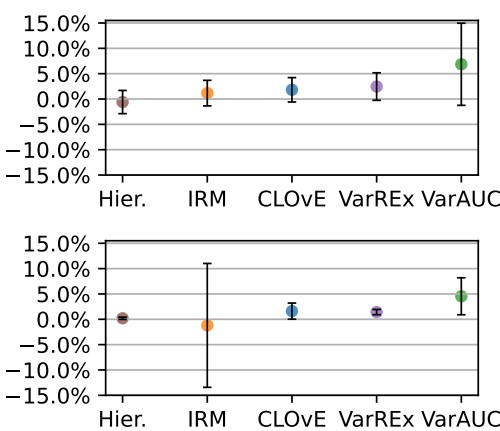

**Experiment 2 - Face Recognition** We used the *CelebA* dataset [30] which contains 202,599 images of 10,177 celebrities. We filtered out people for which the dataset contains less than three images. Following Vinyals et al. [51], we implemented $g$ as a convolutional neural network which has four modules with $3 \times 3$ convolutions and 64 filters, followed by batch normalization, a ReLU activation, $2 \times 2$ max-pooling, and a fully connected layer of size 32. We used the attribute *blond hair* for the class distribution shift: for training, we mainly sampled people without blond hair ($95\%$), while at test time, most people ($95\%$)

Figure 6: Average percentage changes of our method compared to ERM across 10 repetitions are shown for the ETHEC (top) and CelebA (bottom) datasets. Error bars represent $\pm$ one std-dev.

had blond hair. Each training iteration had 150 synthetic environments of two classes and $2r = 20$ data points per class.

We trained the models on 200 synthetic environments at a time, each of two classes. We implemented $g$ as a fully connected neural network with layers of sizes 128, 64, 32 and 16, and ReLU activations between them.

**Experimental Results** As can be seen in Figure 6, while all versions of our algorithm show some improvement over ERM, the best results are achieved with the VarAUC penalty (exact means and standard deviations are reported in Table 3 in Appendix D). One-sided paired t-tests show that the improvement over ERM achieved by our algorithm with the VarAUC penalty is statistically significant, with p-values of $< 0.04$ on both datasets; p-values for other penalties are reported in Table 4. All p-values were adjusted with FDR [4] correction.

In Appendix D we also provide additional analysis confirming that the main improvement of our algorithm over the ERM baseline stems from improved performance on negative minority pairs.

## 6 Discussion

In this study, we examined class distribution shifts in zero-shot learning, with a focus on shifts induced by unknown attributes. Such shifts pose significant challenges in zero-shot learning where new classes emerge in testing, causing standard techniques trained via ERM to fail on shifted class distributions, even when the conditional distribution of the data given class remains the same.

Previous research (see Appendix A) assumes closed-world classification or a known cause, making these methods unsuitable for zero-shot learning or shifts caused by unknown attributes. In response, we introduced a framework and the first algorithm to address class distribution shifts in zero-shot learning using OOD environment balancing methods.

In the causal terminology of closed-world OOD generalization, our framework employs synthetic environments to intervene on attribute mixtures by sampling small class subsets, thereby manipulating the class distribution. This facilitates the creation of diverse environments with varied attribute mixtures, enhancing the distinction between negative examples. A further comparison of our framework with OOD environment balancing methods is provided in Appendix C. Additionally, our proposed VarAUC penalty, designed for metric losses, enhances the separation of negative examples.

Our results demonstrate improvements compared to the ERM baseline on shifted distributions, without compromising performance on unshifted distributions, enabling the learning of more robust representations for zero-shot tasks and ensuring reliable performance.

While the proposed framework is general, our current experiments address shifts in a binary attribute. We defer exploration of additional scenarios, such as those involving shifts in multiple correlated attributes, to future work. An additional promising direction for future work is the consideration of shifts where the responsible attribute is strongly correlated with additional attributes or covariates. This opens up possibilities to explore structured constructions of synthetic environments that leverage such correlations.

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

## A Related Work on Class Distribution Shifts in Closed-World Settings

In *class-imbalanced learning* [28, 10, 8] it is assumed that some classes are more dominant in training, while in deployment this is no longer the case. Therefore, solutions classically include data or loss re-weighting [46, 6, 40, 32] and calibration of the classification score [44, 60]. A popular framework for addressing class distribut ion shifts is *distributionally robust optimization* (DRO) [3, 13, 12, 55], where instead of assuming a specific probability distribution, a set or range of possible distributions is considered, and optimization is performed to achieve the best results on the worst-case distribution. A special case known as *group DRO* [45, 38], involves a group variable that introduces discriminatory patterns among classes within specific groups. The framework to address this includes methods that assume that the classifier does not have access to the group information, and therefore propose re-weighting high loss examples [29], and data sub-sampling to balance classes and groups [21]. Nevertheless, the methods mentioned above rely on the training and test class sets being identical, making them unsuitable for direct application in zero-shot learning scenarios.

## B Analysis of the Parametric Model

### B.1 Derivation of the Loss

We begin by revisiting the parametric model introduced in §3. Let $z_i | c_i \sim \mathcal{N}(c_i, \Sigma_z)$, where $\Sigma_z = \nu_z I_d$, $0 < \nu_z \in \mathbb{R}$, and $I_d$ is the $d$ dimensional identity matrix. Classes $c_i$ are drawn according to a Gaussian distribution $c_i \sim \mathcal{N}(0, \Sigma_a)$ corresponding to their type $a \in \{a_1, a_2\}$. Here, we use a simpler (although less intuitive) notation for the values of the diagonal matrices $\Sigma_a$:

$$\Sigma_{a_1} = \text{diag}\left(\overbrace{\nu_0, \ldots, \nu_0}^{d_0}, \overbrace{\nu_1, \ldots, \nu_1}^{d_1}, \overbrace{\nu_2, \ldots, \nu_2}^{d_2}\right),$$
$$\Sigma_{a_2} = \text{diag}\left(\nu_0, \ldots, \nu_0, \nu_2, \ldots, \nu_2, \nu_1, \ldots, \nu_1\right),$$

where $0 < \nu_2 < \nu_z < \nu_0 < \nu_1$.

We consider a weight representations $g(z) = Wz$, where $W$ is a diagonal matrix with diagonal $w \in \mathbb{R}^d$.

Since $\Sigma_z$ is of full rank, it suffices to consider the no-hinge version of the contrastive loss, that is

$$\widetilde{\ell}(z_i, z_j, y_{ij}; d_g) := y_{ij} \|W(z_i - z_j)\|^4 + (1 - y_{ij})\left(m - \|W(z_i - z_j)\|^2\right)^2, \qquad (12)$$

where $d_g(z_i, z_j) := \|g(z_i - z_j)\|^2 = \|W(z_i - z_j)\|^2$ ($\|\cdot\|$ denotes the Euclidean norm[4]).

For a balanced sample of positive and negative examples, the expected loss is given by

$$\mathbb{E}\left[\widetilde{\ell}(z_i, z_j, y_{ij}; d_g)\right] = \frac{1}{2}\mathbb{E}_{y_{ij}=1}\left[\|W(z_i - z_j)\|^4\right]$$
$$+ \frac{1}{2}\mathbb{E}_{y_{ij}=0}\left[m^2 - 2m\|W(z_i - z_j)\|^2 + \|W(z_i - z_j)\|^4\right]. \qquad (13)$$

To calculate the expression above, we begin by proving the following lemma:

**Lemma 1.** *Let $\mu \in \mathbb{R}^d$ be a random variable and let $t|\mu \sim \mathcal{N}(\mu, \Sigma)$. If $\mu \equiv 0$ (constant), then*

1. $\mathbb{E}\|t\|^4 = 2\,\text{tr}(\Sigma^2) + \text{tr}^2(\Sigma)$.

*If $\mu \sim \mathcal{N}(0, \Sigma_\mu)$, then*

2. $\mathbb{E}\|t\|^2 = \text{tr}(\Sigma) + \text{tr}(\Sigma_\mu)$,

3. $\mathbb{E}\|t\|^4 = 2\,\text{tr}(\Sigma^2) + 4\,\text{tr}(\Sigma\Sigma_\mu) + \text{tr}^2(\Sigma) + 2\,\text{tr}(\Sigma)\,\text{tr}(\Sigma_\mu) + 2\,\text{tr}(\Sigma_\mu^2) + \text{tr}^2(\Sigma_\mu)$.

---

[4]Squared distance is selected for its simplicity in computing the expected value of even powers of the Euclidean norm of Gaussian variables.

*Proof.* For any random variable $u \in \mathbb{R}^d$, such that $u \sim \mathcal{N}(\mu_u, \Sigma_u)$, and any symmetric matrix $A$, we have

$$\mathbb{E}_u[u^T A u] = \mathrm{tr}(A\Sigma_u) + \mu_u^T A \mu_u, \tag{14}$$

$$\mathbb{E}_u[u^T A u]^2 = 2\,\mathrm{tr}\left((A\Sigma_u)^2\right) + 4\mu_u^T A \Sigma_u A \mu_u + \left(\mathrm{tr}(A\Sigma_u) + \mu_u^T A \mu_u\right)^2 \tag{15}$$

(see, for example, Thm. 3.2b.2 in [31]).

First, letting $\mu_u = 0$, $\Sigma_u = \Sigma$ and $A = I_d$ in (15) we get

$$\mathbb{E}\,\|t\|^4 = \mathbb{E}[t^T t]^2 = 2\,\mathrm{tr}(\Sigma^2) + \mathrm{tr}^2(\Sigma). \tag{16}$$

Now, assume that $\mu \sim \mathcal{N}(0, \Sigma_\mu)$. From (14) we get $\mathbb{E}_\mu\,\|\mu\|^2 = \mathbb{E}_\mu[\mu^T \mu] = \mathrm{tr}(\Sigma_\mu)$, and thus

$$\mathbb{E}\,\|t\|^2 = \mathbb{E}_\mu\left[\mathbb{E}_{t|\mu}[t^T t \mid \mu]\right] = \mathbb{E}_\mu\left[\mathrm{tr}(\Sigma) + \mu^T \mu\right] = \mathrm{tr}(\Sigma) + \mathrm{tr}(\Sigma_\mu). \tag{17}$$

Similarly, from (15) we have

$$\mathbb{E}\,\|t\|^4 = \mathbb{E}_\mu\left[\mathbb{E}_{t|\mu}[[t^T t]^2 \mid \mu]\right] = 2\,\mathrm{tr}(\Sigma^2) + 4\,\mathbb{E}_\mu[\mu^T \Sigma \mu] + \mathrm{tr}^2(\Sigma) + 2\,\mathrm{tr}(\Sigma)\,\mathbb{E}_\mu\,\|\mu\|^2 + \mathbb{E}_\mu\,\|\mu\|^4. \tag{18}$$

By substituting $A = \Sigma$ in (14) we get $\mathbb{E}_\mu[\mu^T \Sigma \mu] = \mathrm{tr}(\Sigma\Sigma_\mu)$, and from (15) we have

$$\mathbb{E}_\mu\,\|\mu\|^4 = 2\,\mathrm{tr}(\Sigma_\mu^2) + \mathrm{tr}^2(\Sigma_\mu). \tag{19}$$

Therefore,

$$\mathbb{E}\,\|t\|^4 = 2\,\mathrm{tr}(\Sigma^2) + 4\,\mathrm{tr}(\Sigma\Sigma_\mu) + \mathrm{tr}^2(\Sigma) + 2\,\mathrm{tr}(\Sigma)\,\mathrm{tr}(\Sigma_\mu) + 2\,\mathrm{tr}(\Sigma_\mu^2) + \mathrm{tr}^2(\Sigma_\mu). \tag{20}$$

$\square$

Note that $W(z_i - z_j) \sim \mathcal{N}(\mu, \Sigma)$, with $\mu = W(c_i - c_j)$ and $\Sigma = 2\nu_z W^T W$.

If $y_{ij} = 1$, then $z_i$ and $z_j$ are from the same class, meaning that $c_i = c_j$ and thus $\mu = 0$. Therefore, by Lemma 1.(1) we have

$$\begin{aligned}
\mathbb{E}_{y_{ij}=1}\,\|W(z_i - z_j)\|^4 &= 2\,\mathrm{tr}(\Sigma^2) + \mathrm{tr}^2(\Sigma) \\
&= 2 \cdot 4\nu_z^2\,\mathrm{tr}\left(\left[W^T W\right]^2\right) + 4\nu_z^2\,\mathrm{tr}^2(W^T W) \\
&= 8\nu_z^2 \sum_{i=1}^d w_i^4 + 4\nu_z^2 \left(\sum_{i=1}^d w_i^2\right)^2.
\end{aligned} \tag{21}$$

However, for pairs from different classes, that is, when $y_{ij} = 0$, the mean $\mu$ is itself a Gaussian random variable distributed according to $\mathcal{N}(0, \Sigma_\mu)$, where

$$\Sigma_\mu = \begin{cases} W^T (2\Sigma_{a_1}) W & c_i, c_j \text{ are both of type } a_1 \\ W^T (2\Sigma_{a_2}) W & c_i, c_j \text{ are both of type } a_2 \\ W^T (\Sigma_{a_1} + \Sigma_{a_2}) W & c_i, c_j \text{ are of different types .} \end{cases} \tag{22}$$

Therefore, by Lemma 1.(2) we have

$$\begin{aligned}
\mathbb{E}_{y_{ij}=0}\,\|W(z_i - z_j)\|^2 &= \mathbb{E}_{y_{ij}=0}\left[\mathrm{tr}(\Sigma_\mu) + \mathrm{tr}(\Sigma)\right] = \mathbb{E}_{y_{ij}=0}\left[\mathrm{tr}(\Sigma_\mu)\right] + \mathrm{tr}(\Sigma) \\
&= \rho^2\,\mathrm{tr}\left(2W^T \Sigma_{a_1} W\right) + (1-\rho)^2\,\mathrm{tr}\left(2W^T \Sigma_{a_2} W\right) \\
&\quad + 2\rho(1-\rho)\,\mathrm{tr}\left(W^T (\Sigma_{a_1} + \Sigma_{a_2}) W\right) + \mathrm{tr}(\Sigma) \\
&= 2\left[(\nu_0 + \nu_z) \sum_{i=1}^{d_0} w_i^2 + (\alpha_1 + \nu_z) \sum_{i=d_0+1}^{d_0+d_1} w_i^2 + (\alpha_2 + \nu_z) \sum_{i=d_0+d_1+1}^{d} w_i^2\right],
\end{aligned} \tag{23}$$

where

$$\alpha_1 := \rho^2 \nu_1 + (1-\rho)^2 \nu_2 + \rho(1-\rho)\left(\nu_1 + \nu_2\right) = \rho\nu_1 + (1-\rho)\,\nu_2,$$
$$\alpha_2 := \rho^2 \nu_2 + (1-\rho)^2 \nu_1 + \rho(1-\rho)\left(\nu_1 + \nu_2\right) = \rho\nu_2 + (1-\rho)\,\nu_1,$$
$$\beta_1 := 2\rho^2 \nu_1^2 + 2(1-\rho)^2 \nu_2^2 + \rho(1-\rho)\left(\nu_1 + \nu_2\right)^2,$$
$$\beta_2 := 2\rho^2 \nu_2^2 + 2(1-\rho)^2 \nu_1^2 + \rho(1-\rho)\left(\nu_1 + \nu_2\right)^2. \tag{24}$$

By Lemma 1.(3) we have

$$
\begin{aligned}
\mathbb{E}_{y_{ij}=0}\left\| w\left(z_i - z_j\right)\right\|^4 &= \mathbb{E}_{y_{ij}=0}\Big[2\operatorname{tr}\left(\Sigma^2\right) + 4\operatorname{tr}\left(\Sigma\Sigma_\mu\right) + \operatorname{tr}^2(\Sigma) + 2\operatorname{tr}\left(\Sigma\right)\operatorname{tr}\left(\Sigma_\mu\right) \\
&\quad + 2\operatorname{tr}\left(\Sigma_\mu^2\right) + \left(\operatorname{tr}\left(\Sigma_\mu\right)\right)^2\Big] \\
&= 2\operatorname{tr}(\Sigma^2) + 4\,\mathbb{E}_{y_{ij}=0}[\operatorname{tr}(\Sigma\Sigma_\mu)] + \operatorname{tr}^2(\Sigma) + 2\operatorname{tr}(\Sigma)\,\mathbb{E}_{y_{ij}=0}[\operatorname{tr}(\Sigma_\mu)] \\
&\quad + 2\,\mathbb{E}_{y_{ij}=0}[\operatorname{tr}(\Sigma_\mu^2)] + \mathbb{E}_{y_{ij}=0}[\operatorname{tr}^2(\Sigma_\mu)],
\end{aligned}
\tag{25}
$$

where

$$
\begin{aligned}
\mathbb{E}_{y_{ij}=0}\left[\operatorname{tr}\left(\Sigma\Sigma_\mu\right)\right] &= 2\nu_z\operatorname{tr}\Big(W^T W\left[2\rho^2 W^T \Sigma_{a_1} W + 2(1-\rho)^2 W^T \Sigma_{a_2} W\right. \\
&\quad \left. + 2\rho(1-\rho)W^T\left(\Sigma_{a_1} + \Sigma_{a_2}\right)W\right]\Big) \\
&= 4\nu_z\left[\nu_0 \sum_{i=1}^{d_0} w_i^4 + \alpha_1 \sum_{i=d_0+1}^{d_0+d_1} w_i^4 + \alpha_2 \sum_{i=d_0+d_1+1}^{d} w_i^4\right];
\end{aligned}
\tag{26}
$$

$$
\begin{aligned}
\mathbb{E}_{y_{ij}=0}\left[\operatorname{tr}\left(\Sigma_\mu\right)\right] &= \rho^2 \operatorname{tr}\left(2W^T \Sigma_{a_1} W\right) + (1-\rho)^2 \operatorname{tr}\left(2W^T \Sigma_{a_1} W\right) \\
&\quad + 2\rho(1-\rho)\operatorname{tr}\left(W^T\left(\Sigma_{a_1} + \Sigma_{a_2}\right)W\right) \\
&= 2\left[\nu_0 \sum_{i=1}^{d_0} w_i^2 + \alpha_1 \sum_{i=d_0+1}^{d_0+d_1} w_i^2 + \alpha_2 \sum_{i=d_0+d_1+1}^{d} w_i^2\right], r
\end{aligned}
\tag{27}
$$

and so

$$\operatorname{tr}\left(\Sigma\right)\mathbb{E}_{y_{ij}=0}\left[\operatorname{tr}\left(\Sigma_\mu\right)\right] = 4\nu_z\left(\sum_{i=1}^{d} w_i^2\right)\left[\nu_0 \sum_{i=1}^{d_0} w_i^2 + \alpha_1 \sum_{i=d_0+1}^{d_0+d_1} w_i^2 + \alpha_2 \sum_{i=d_0+d_1+1}^{d} w_i^2\right]; \tag{28}$$

$$
\begin{aligned}
\mathbb{E}_{y_{ij}=0}\left[\operatorname{tr}\left(\Sigma_\mu^2\right)\right] &= \rho^2 \operatorname{tr}\left((2W^T \Sigma_{a_1} W)^2\right) + (1-\rho)^2 \operatorname{tr}\left((2W^T \Sigma_{a_2} W)^2\right) \\
&\quad + 2\rho(1-\rho)\operatorname{tr}\left((W^T\left(\Sigma_{a_1} + \Sigma_{a_2}\right)W)^2\right) \\
&= 2\left[2\nu_0^2 \sum_{i=1}^{d_0} w_i^4 + \beta_1 \sum_{i=d_0+1}^{d_0+d_1} w_i^4 + \beta_2 \sum_{i=d_0+d_1+1}^{d} w_i^4\right];
\end{aligned}
\tag{29}
$$

and similarly

$$
\begin{aligned}
\mathbb{E}_{y_{ij}=0}\left[\operatorname{tr}^2\left(\Sigma_\mu\right)\right] &= 2\Bigg[2\nu_0^2\left(\sum_{i=1}^{d_0} w_i^2\right)^2 + \beta_1\left(\sum_{i=d_0+1}^{d_0+d_1} w_i^2\right)^2 + \beta_2\left(\sum_{i=d_0+d_1+1}^{d} w_i^2\right)^2 \\
&\quad + 4\gamma_{0,1}\sum_{i=1}^{d_0} w_i^2 \sum_{i=d_0+1}^{d_0+d_1} w_i^2 + 4\gamma_{0,2}\sum_{i=1}^{d_0} w_i^2 \sum_{i=d_0+d_1+1}^{d} w_i^2 \\
&\quad + 4\gamma_{1,2}\sum_{i=d_0+1}^{d_0+d_1} w_i^2 \sum_{i=d_0+d_1+1}^{d} w_i^2\Bigg],
\end{aligned}
\tag{30}
$$

where we denote for short

$$\gamma_{0,1} := \rho^2 \nu_0 \nu_1 + (1-\rho)^2 \nu_0 \nu_2 + \rho(1-\rho)\nu_0\left(\nu_1 + \nu_2\right),$$
$$\gamma_{0,2} := \rho^2 \nu_0 \nu_2 + (1-\rho)^2 \nu_0 \nu_1 + \rho(1-\rho)\nu_0\left(\nu_1 + \nu_2\right),$$
$$\gamma_{1,2} := \rho^2 \nu_1 \nu_2 + (1-\rho)^2 \nu_1 \nu_2 + \frac{1}{2}\rho(1-\rho)\left(\nu_1 + \nu_2\right)^2. \tag{31}$$

Finally, due to symmetry, at the optimal solution we have

$$w_i = \begin{cases} u_0 & 0 \le i \le d_0 \\ u_1 & d_0 + 1 \le i \le d_0 + d_1 \\ u_2 & d_0 + d_1 + 1 \le i \le d, \end{cases} \tag{32}$$

and by combining these results, we get

$$\begin{aligned}
\mathbb{E}\left[\widetilde{\ell}\left(z_i, z_j, y_{ij}; d_g\right)\right] = {} & d_0 u_0^4 \left(8\nu_z^2 + 8\nu_z \nu_0 + 4\nu_0^2 + 2\nu_0^2 d_0\right) \\
& + d_1 u_1^4 \left(8\nu_z^2 + 8\nu_z \alpha_1 + 2\beta_1 + \beta_1 d_1\right) \\
& + d_2 u_2^4 \left(8\nu_z^2 + 8\nu_z \alpha_2 + 2\beta_2 + \beta_2 d_2\right) \\
& - 2d_0 u_0^2 \left(\nu_0 + \nu_z\right) - 2d_1 u_1^2 \left(\alpha_1 + \nu_z\right) - 2d_2 u_2^2 \left(\alpha_2 + \nu_z\right) \\
& + 4\nu_z^2 \left(d_0 u_0^2 + d_1 u_1^2 + d_2 u_2^2\right)^2 + \frac{1}{2}m \\
& + 4\nu_z \left(d_0 u_0^2 + d_1 u_1^2 + d_2 u_2^2\right)\left[\nu_0 d_0 u_0^2 + \alpha_1 d_1 u_1^2 + \alpha_2 d_2 u_2^2\right] \\
& + 4\gamma_{0,1} d_0 d_1 u_0^2 u_1^2 + 4\gamma_{0,2} d_0 d_2 u_0^2 u_2^2 + 4\gamma_{1,2} d_1 d_2 u_1^2 u_2^2. \tag{33}
\end{aligned}$$

## B.2 Analysis of the Optimal Solution (Proof of Proposition 1)

Proposition 1 shows that when $d_1$ and $d_2$ are relatively similar, the optimal solution on the training distribution, assigns more weight to components with high variance in the training data than to those with high variance in the shifted test distribution.

We begin by defining the required condition on $d_1$ and $d_2$. Denote

$$\begin{aligned}
\psi_1 &:= 2\nu_z^2 + 2\nu_z \alpha_1 + \beta_1 \\
\psi_2 &:= 2\nu_z^2 + 2\nu_z \alpha_2 + \beta_2 \\
\eta_{01} &:= 4\nu_z^2 + 2\nu_z \left(\alpha_1 + \nu_0\right) + 2\gamma_{0,1} \\
\eta_{02} &:= 4\nu_z^2 + 2\nu_z \left(\alpha_2 + \nu_0\right) + 2\gamma_{0,2} \\
\eta_{12} &:= 4\nu_z^2 + 2\nu_z \left(\alpha_1 + \alpha_2\right) + 2\gamma_{1,2}. \tag{34}
\end{aligned}$$

Then for $\alpha, \beta, \gamma$ values as in equations 24 and 31, we define

$$h_l\left(\rho, \nu_z, \nu_0, \nu_1, \nu_2\right) := \frac{\psi_1}{\psi_2}\frac{\left(\alpha_2 + \nu_z\right)}{\left(\alpha_1 + \nu_z\right)}, \quad h_u\left(\rho, \nu_z, \nu_0, \nu_1, \nu_2\right) := \frac{\psi_1}{\psi_2}\frac{\eta_{02}}{\eta_{01}}, \tag{35}$$

and the corresponding condition

$$h_l\left(\rho, \nu_z, \nu_0, \nu_1, \nu_2\right) < \frac{d_2 + 2}{d_1 + 2} < h_u\left(\rho, \nu_z, \nu_0, \nu_1, \nu_2\right). \tag{36}$$

While this condition is sufficient, it is not necessary. Values of $\rho, \nu_z, \nu_0, \nu_1, \nu_2$ and $d_1, d_2$ that satisfy 35 provide an example requiring only a simple analysis, without a full characterization of the optimal solution, for the failure of optimization over the training distribution. However, such failures can occur for additional parameter values, and the full characterization is provided in Appendix B.3.

*Proof.* Without loss of generality assume m=1. Then,

$$\frac{\partial \mathbb{E}\left[\widetilde{\ell}\left(z_i, z_j, y_{ij}; d_g\right)\right]}{\partial u_0^2} = 2d_0 u_0^2 \left(8\nu_z^2 + 8\nu_z\nu_0 + 4\nu_0^2 + 2\nu_0^2 d_0\right)$$

$$- 2d_0 \left(\nu_0 + \nu_z\right)$$

$$+ 8d_0\nu_z^2 \left(d_0 u_0^2 + d_1 u_1^2 + d_2 u_2^2\right)$$

$$+ 4d_0\nu_z \left(\nu_0 d_0 u_0^2 + \alpha_1 d_1 u_1^2 + \alpha_2 d_2 u_2^2\right)$$

$$+ 4d_0\nu_z\nu_0 \left(d_0 u_0^2 + d_1 u_1^2 + d_2 u_2^2\right)$$

$$+ 4\gamma_{0,1} d_0 d_1 u_1^2 + 4\gamma_{0,2} d_0 d_2 u_2^2 \tag{37}$$

and by setting the partial derivative to zero we get

$$2u_0^2 \left(2 + d_0\right)\left(2\nu_z^2 + 2\nu_z\nu_0 + \nu_0^2\right) = 2d_1 u_1^2 \left(2\nu_z^2 + \nu_z\left(\alpha_1 + \nu_0\right) + \gamma_{0,1}\right)$$

$$+ 2d_2 u_2^2 \left(2\nu_z^2 + \nu_z\left(\alpha_2 + \nu_0\right) + \gamma_{0,2}\right) - \left(\nu_0 + \nu_z\right). \tag{38}$$

Therefore,

$$u_0^2 = \frac{\nu_0 + \nu_z - \eta_{01} d_1 u_1^2 - \eta_{02} d_2 u_2^2}{2\left(2 + d_0\right)\left(2\nu_z^2 + 2\nu_z\nu_0 + \nu_0^2\right)}. \tag{39}$$

and similarly

$$u_1^2 = \frac{\left(\alpha_1 + \nu_z\right) - \eta_{01} d_0 u_0^2 - \eta_{12} d_2 u_2^2}{2\left(2 + d_1\right)\left(2\nu_z^2 + 2\nu_z\alpha_1 + \beta_1\right)} \tag{40}$$

$$u_2^2 = \frac{\left(\alpha_2 + \nu_z\right) - \eta_{02} d_0 u_0^2 - \eta_{12} d_1 u_1^2}{2\left(2 + d_2\right)\left(2\nu_z^2 + 2\nu_z\alpha_2 + \beta_2\right)}. \tag{41}$$

Hence,

$$d_1 u_1^2 - d_2 u_2^2 = \frac{\left(2 + d_2\right)\left(2\nu_z^2 + 2\nu_z\alpha_2 + \beta_2\right)\left[d_1\left(\alpha_1 + \nu_z\right) - \eta_{01} d_1 d_0 u_0^2 - \eta_{12} d_1 d_2 u_2^2\right]}{2\left(2 + d_1\right)\left(2 + d_2\right)\left(2\nu_z^2 + 2\nu_z\alpha_1 + \beta_1\right)\left(2\nu_z^2 + 2\nu_z\alpha_2 + \beta_2\right)}$$

$$- \frac{\left(2 + d_1\right)\left(2\nu_z^2 + 2\nu_z\alpha_1 + \beta_1\right)\left[d_2\left(\alpha_2 + \nu_z\right) - \eta_{02} d_2 d_0 u_0^2 - \eta_{12} d_1 d_2 u_1^2\right]}{2\left(2 + d_1\right)\left(2 + d_2\right)\left(2\nu_z^2 + 2\nu_z\alpha_1 + \beta_1\right)\left(2\nu_z^2 + 2\nu_z\alpha_2 + \beta_2\right)}. \tag{42}$$

Denoting

$$\xi := 2\left(2 + d_1\right)\left(2 + d_2\right)\left(2\nu_z^2 + 2\nu_z\alpha_1 + \beta_1\right)\left(2\nu_z^2 + 2\nu_z\alpha_2 + \beta_2\right)$$

$$= 2\left(2 + d_1\right)\left(2 + d_2\right)\psi_1\psi_2$$

we have

$$d_1 u_1^2 \left[1 - \frac{1}{\xi}\left(2 + d_1\right)\psi_1\eta_{12}\right] = d_2 u_2^2 \left[1 - \frac{1}{\xi}\left(2 + d_2\right)\psi_2\eta_{12}\right]$$

$$+ \frac{1}{\xi}\left(2 + d_2\right)\psi_2\left(\alpha_1 + \nu_z\right) - \frac{1}{\xi}\left(2 + d_1\right)\psi_1\left(\alpha_2 + \nu_z\right)$$

$$+ d_0 u_0^2 \left[\frac{1}{\xi}\left(2 + d_1\right)\psi_1\eta_{02} - \frac{1}{\xi}\left(2 + d_2\right)\psi_2\eta_{01}\right]$$

and therefore

$$d_1 u_1^2 - d_2 u_2^2 = d_2 u_2^2 \left(\frac{1 - \frac{1}{\xi}\left(2 + d_2\right)\psi_2\eta_{12}}{1 - \frac{1}{\xi}\left(2 + d_1\right)\psi_1\eta_{12}} - 1\right)$$

$$+ \frac{1}{2\left(2 + d_1\right)\left(2 + d_2\right)\psi_1\psi_2}\frac{\left(2 + d_2\right)\psi_2\left(\alpha_1 + \nu_z\right) - \left(2 + d_1\right)\psi_1\left(\alpha_2 + \nu_z\right)}{1 - \frac{1}{\xi}\left(2 + d_1\right)\psi_1\eta_{12}}$$

$$+ d_0 u_0^2 \frac{1}{2\left(2 + d_1\right)\left(2 + d_2\right)\psi_1\psi_2}\left[\frac{\left(2 + d_1\right)\psi_1\eta_{02}}{1 - \frac{1}{\xi}\left(2 + d_1\right)\psi_1\eta_{12}} - \frac{\left(2 + d_2\right)\psi_2\eta_{01}}{1 - \frac{1}{\xi}\left(2 + d_1\right)\psi_1\eta_{12}}\right]. \tag{43}$$

Denote

$$\Delta = (2 + d_1) \left[ d_2 u_2^2 \eta_{12} \psi_1 - (\alpha_2 + \nu_z) \psi_1 + d_0 u_0^2 \psi_1 \eta_{02} \right]$$
$$- (2 + d_2) \left[ d_2 u_2^2 \eta_{12} \psi_2 - (\alpha_1 + \nu_z) \psi_2 + d_0 u_0^2 \psi_2 \eta_{01} \right], \quad (44)$$

and thus

$$d_1 u_1^2 - d_2 u_2^2 = \frac{1}{2(2 + d_1)(2 + d_2)\psi_1\psi_2} \frac{1}{1 - \frac{1}{\xi}(2 + d_1)\psi_1\eta_{12}} \Delta. \quad (45)$$

Note that for $d_1, d_2$ such that

$$\begin{cases} (2 + d_1)\psi_1 - (2 + d_2)\psi_2 > 0 & \Rightarrow \frac{d_2 + 2}{d_1 + 2} < \frac{\psi_1}{\psi_2} \\ (2 + d_2)(\alpha_1 + \nu_z)\psi_2 - (2 + d_1)(\alpha_2 + \nu_z)\psi_1 > 0 & \Rightarrow \frac{d_2 + 2}{d_1 + 2} > \frac{\psi_1}{\psi_2}\frac{(\alpha_2 + \nu_z)}{(\alpha_1 + \nu_z)} \\ (2 + d_1)\psi_1\eta_{02} - (2 + d_2)\psi_2\eta_{01} > 0 & \Rightarrow \frac{d_2 + 2}{d_1 + 2} < \frac{\psi_1}{\psi_2}\frac{\eta_{02}}{\eta_{01}} \end{cases}$$

we have $\Delta > 0$. Since $\frac{\eta_{02}}{\eta_{01}} < 1$, this reduces to the last two conditions and therefore, in particular for

$$\frac{\psi_1}{\psi_2}\frac{(\alpha_2 + \nu_z)}{(\alpha_1 + \nu_z)} < \frac{d_2 + 2}{d_1 + 2} < \frac{\psi_1}{\psi_2}\frac{\eta_{02}}{\eta_{01}} \quad (46)$$

we have $\Delta > 0$. Additionally, note that

$$1 - \frac{1}{\xi}(2 + d_1)\psi_1\eta_{12} = 1 - \frac{\eta_{12}}{2(2 + d_1)(2 + d_2)\psi_1\psi_2}(2 + d_1)\psi_1 = \frac{2(2 + d_2)\psi_2 - \eta_{12}}{2(2 + d_2)\psi_2} \quad (47)$$

and thus $1 - \frac{1}{\xi}(2 + d_1)\psi_1\eta_{12} > 0$ iff

$$d_2 + 2 > \frac{1}{2}\frac{\eta_{12}}{\psi_2}. \quad (48)$$

Combining these conditions reduces to

$$\frac{\psi_1}{\psi_2}\frac{(\alpha_2 + \nu_z)}{(\alpha_1 + \nu_z)} < \frac{d_2 + 2}{d_1 + 2} < \frac{\psi_1}{\psi_2}\frac{\eta_{02}}{\eta_{01}}, \quad (49)$$

and therefore, for $\nu_z, \nu_0, \nu_1, \nu_2, d_1, d_2$ satisfying

$$\frac{\psi_1}{\psi_2}\frac{(\alpha_2 + \nu_z)}{(\alpha_1 + \nu_z)} < \frac{d_2 + 2}{d_1 + 2} < \frac{\psi_1}{\psi_2}\frac{\eta_{02}}{\eta_{01}}. \quad (50)$$

we have $d_1 u_1^2 - d_2 u_2^2 > 0$.[5]                                                          □

---

[5]Similarly, the condition obtained for $1 - \frac{1}{\xi}(2 + d_1)\psi_1\eta_{12} < 0$ and $\Delta < 0$ is $\frac{\psi_1}{\psi_2}(2 + d_1) < 2 + d_2 < \frac{\psi_1}{\psi_2}\frac{(\alpha_2 + \nu_z)}{(\alpha_1 + \nu_z)}(2 + d_1)$, which cannot be achieved since $\alpha_2 < \alpha_1$.

## B.3  Explicit Expression for the Optimal Representation

In order to derive the optimal representation, we differentiate the expected loss with respect to the squared values in the diagonal of $W$, that is, $w_i^2$:

$$\frac{\partial}{\partial\left(w_i^2\right)}\operatorname{tr}\left(\Sigma^2\right)=8\nu_z^2 w_i^2 \tag{51}$$

$$\frac{\partial}{\partial\left(w_i^2\right)}\operatorname{tr}^2\left(\Sigma\right)=8\nu_z^2\sum_{j=1}^{d}w_j^2 \tag{52}$$

$$\frac{\partial}{\partial\left(w_i^2\right)}\mathbb{E}_{y=0}\left[\operatorname{tr}\left(\Sigma\Sigma_\mu\right)\right]=\begin{cases}8\nu_z\nu_0 w_i^2, & 1\le i\le d_0\\ 8\nu_z\alpha_1 w_i^2, & d_0+1\le i\le d_0+d_1\\ 8\nu_z\alpha_2 w_i^2, & d_0+d_1+1\le i\le d\end{cases} \tag{53}$$

$$\frac{\partial\left[\operatorname{tr}(\Sigma)\,\mathbb{E}_{y=0}[\operatorname{tr}\Sigma_\mu]\right]}{\partial\left(w_i^2\right)}= \tag{54}$$

$$\begin{cases}4\nu_z\left[2\nu_0\sum_{j=1}^{d_0}w_j^2+(\alpha_1+\nu_0)\sum_{j=d_0+1}^{d_0+d_1}w_j^2+(\alpha_2+\nu_0)\sum_{j=d_0+d_1+1}^{d}w_j^2\right] & 1\le i\le d_0\\[3mm] 4\nu_z\left[(\nu_0+\alpha_1)\sum_{j=1}^{d_0}w_j^2+2\alpha_1\sum_{j=d_0+1}^{d_0+d_1}w_j^2+(\alpha_2+\alpha_1)\sum_{j=d_0+d_1+1}^{d}w_j^2\right] & d_0+1\le i\le d_0+d_1\\[3mm] 4\nu_z\left[(\nu_0+\alpha_2)\sum_{j=1}^{d_0}w_j^2+(\alpha_1+\alpha_2)\sum_{j=d_0+1}^{d_0+d_1}w_j^2+2\alpha_2\sum_{j=d_0+d_1+1}^{d}w_j^2\right] & d_0+d_1+1\le i\le d\end{cases} \tag{55}$$

$$\frac{\partial}{\partial\left(w_i^2\right)}\mathbb{E}_{y=0}\left[\operatorname{tr}\left(\Sigma_\mu^2\right)\right]=\begin{cases}8\nu_0^2 w_i^2 & 1\le i\le d_0\\ 4\beta_1 w_i^2 & d_0+1\le i\le d_0+d_1\\ 4\beta_2 w_i^2 & d_0+d_1+1\le i\le d\end{cases} \tag{56}$$

$$\frac{\partial}{\partial\left(w_i^2\right)}\mathbb{E}_{y=0}\left[\operatorname{tr}^2\left(\Sigma_\mu\right)\right]= \tag{57}$$

$$\begin{cases}8\nu_0^2\sum_{j=1}^{d_0}w_j^2+8\gamma_{0,1}\sum_{j=d_0+1}^{d_0+d_1}w_j^2+8\gamma_{0,2}\sum_{j=d_0+d_1+1}^{d}w_j^2 & 1\le i\le d_0\\[3mm] 8\gamma_{0,1}\sum_{j=1}^{d_0}w_j^2+4\beta_1\sum_{j=d_0+1}^{d_0+d_1}w_j^2+8\gamma_{1,2}\sum_{j=d_0+d_1+1}^{d}w_j^2 & d_0+1\le i\le d_0+d_1\\[3mm] 8\gamma_{0,2}\sum_{j=1}^{d_0}w_j^2+8\gamma_{1,2}\sum_{j=d_0+1}^{d_0+d_1}w_j^2+4\beta_2\sum_{j=d_0+d_1+1}^{d}w_j^2 & d_0+d_1+1\le i\le d\end{cases} \tag{58}$$

Combining these results, we get for $1\le i\le d_0$

$$\partial_0:=\frac{\partial}{\partial\left(w_i^2\right)}\widetilde{\ell}\left(z_i,z_j,y_{ij};d_g\right)=\frac{1}{2}\left[2\cdot 8\nu_z^2 w_i^2+8\nu_z^2\sum_{j=1}^{d}w_j^2\right]-m\left[2\left(\nu_0+\nu_z\right)\right]$$

$$+8\nu_z^2 w_i^2+4\nu_z^2\sum_{j=1}^{d}w_j^2+2\cdot 8\nu_z\nu_0 w_i^2$$

$$+4\nu_z\left[2\nu_0\sum_{j=1}^{d_0}w_j^2+(\alpha_1+\nu_0)\sum_{j=d_0+1}^{d_0+d_1}w_j^2+(\alpha_2+\nu_0)\sum_{j=d_0+d_1+1}^{d}w_j^2\right]$$

$$+8\nu_0^2 w_i^2+\frac{1}{2}\left[8\nu_0^2\sum_{j=1}^{d_0}w_j^2+8\gamma_{0,1}\sum_{j=d_0+1}^{d_0+d_1}w_j^2+8\gamma_{0,2}\sum_{j=d_0+d_1+1}^{d}w_j^2\right],$$

for $d_0 + 1 \leq i \leq d_0 + d_1$

$$\partial_1 := \frac{\partial}{\partial\left(w_i^2\right)} \widetilde{\ell}\left(z_i, z_j, y_{ij}; d_g\right) = \frac{1}{2}\left[2 \cdot 8\nu_z^2 w_i^2 + 8\nu_z^2 \sum_{j=1}^{d} w_j^2\right] - m\left[2\left(\alpha_1 + \nu_z\right)\right]$$

$$+ 8\nu_z^2 w_i^2 + 4\nu_z^2 \sum_{j=1}^{d} w_j^2 + 2 \cdot 8\nu_z \alpha_1 w_i^2$$

$$+ 4\nu_z\left[\left(\nu_0 + \alpha_1\right)\sum_{j=1}^{d_0} w_j^2 + 2\alpha_1 \sum_{j=d_0+1}^{d_0+d_1} w_j^2 + \left(\alpha_2 + \alpha_1\right)\sum_{j=d_0+d_1+1}^{d} w_j^2\right]$$

$$+ 4\beta_1 w_i^2 + \frac{1}{2}\left[8\gamma_{0,1} \sum_{j=1}^{d_0} w_j^2 + 4\beta_1 \sum_{j=d_0+1}^{d_0+d_1} w_j^2 + 8\gamma_{1,2} \sum_{j=d_0+d_1+1}^{d} w_j^2\right],$$

and similarly for $d_0 + d_1 + 1 \leq i \leq d$

$$\partial_2 := \frac{\partial}{\partial\left(w_i^2\right)} \widetilde{\ell}\left(z_i, z_j, y_{ij}; d_g\right) = \frac{1}{2}\left[2 \cdot 8\nu_z^2 w_i^2 + 8\nu_z^2 \sum_{j=1}^{d} w_j^2\right] - m\left[2\left(\alpha_2 + \nu_z\right)\right]$$

$$+ 8\nu_z^2 w_i^2 + 4\nu_z^2 \sum_{j=1}^{d} w_j^2 + 2 \cdot 8\nu_z \alpha_2 w_i^2$$

$$+ 4\nu_z\left[\left(\nu_0 + \alpha_2\right)\sum_{j=1}^{d_0} w_j^2 + \left(\alpha_1 + \alpha_2\right)\sum_{j=d_0+1}^{d_0+d_1} w_j^2 + 2\alpha_2 \sum_{j=d_0+d_1+1}^{d} w_j^2\right]$$

$$+ 4\beta_2 w_i^2 + \frac{1}{2}\left[8\gamma_{0,2} \sum_{j=1}^{d_0} w_j^2 + 8\gamma_{1,2} \sum_{j=d_0+1}^{d_0+d_1} w_j^2 + 4\beta_2 \sum_{j=d_0+d_1+1}^{d} w_j^2\right].$$

Thus, we can write for the symmetric solution

$$\partial_0 = -2m\left(\nu_0 + \nu_z\right) + u_0^2 G_{0,0} + u_1^2 G_{0,1} + u_2^2 G_{0,2}, \tag{59}$$

$$\partial_1 = -2m\left(\alpha_1 + \nu_z\right) + u_0^2 G_{1,0} + u_1^2 G_{1,1} + u_2^2 G_{1,2}, \tag{60}$$

$$\partial_2 = -2m\left(\alpha_2 + \nu_z\right) + u_0^2 G_{2,0} + u_1^2 G_{2,1} + u_2^2 G_{2,2}, \tag{61}$$

where

$$G_{0,0} = 16\nu_z^2 + 8\nu_z^2 d_0 + 16\nu_z\nu_0 + 8\nu_z\nu_0 d_0 + 8\nu_0^2 + 4\nu_0^2 d_0$$

$$G_{0,1} = 8\nu_z^2 d_1 + 4\nu_z\left(\alpha_1 + \nu_0\right) d_1 + 4\gamma_{0,1} d_1$$

$$G_{0,2} = 8\nu_z^2 d_2 + 4\nu_z\left(\alpha_2 + \nu_0\right) d_2 + 4\gamma_{0,2} d_2$$

$$G_{1,0} = 8\nu_z^2 d_0 + 4\nu_z\left(\nu_0 + \alpha_1\right) d_0 + 4\gamma_{0,1} d_0$$

$$G_{1,1} = 16\nu_z^2 + 8\nu_z^2 d_1 + 16\nu_z\alpha_1 + 8\nu_z\alpha_1 d_1 + 4\beta_1 + 4\beta_1 d_1$$

$$G_{1,2} = 8\nu_z^2 d_2 + 4\nu_z\left(\alpha_2 + \alpha_1\right) d_2 + 4\gamma_{1,2} d_2$$

$$G_{2,0} = 8\nu_z^2 d_0 + 4\nu_z\left(\nu_0 + \alpha_2\right) d_0 + 4\gamma_{0,2} d_0$$

$$G_{2,1} = 8\nu_z^2 d_1 + 4\nu_z\left(\alpha_1 + \alpha_2\right) d_1 + 4\gamma_{1,2} d_1$$

$$G_{2,2} = 16\nu_z^2 + 8\nu_z^2 d_2 + 16\nu_z\alpha_2 + 8\nu_z\alpha_2 d_2 + 4\beta_2 + 4\beta_2 d_2.$$

Therefore, the optimal representation is given by the solution to the following set of linear equations:

$$\begin{pmatrix} u_0^2 \\ u_1^2 \\ u_2^2 \end{pmatrix} = 2m\, G^{-1} \begin{pmatrix} \nu_0 + \nu_z \\ \alpha_1 + \nu_z \\ \alpha_2 + \nu_z \end{pmatrix}, \tag{62}$$

Table 1: Simulation results. For each mixture ratio we report the mean AUC and the standard deviation across 10 repetitions of the experiment. Results are reported for in-distribution scenario ($P_C$), and class distribution shift ($Q_C$). Best result is marked in bold.

|  |  | $\rho = 0.05$ | $\rho = 0.1$ | $\rho = 0.3$ |
|---|---|---|---|---|
| In Distribution | ERM | 0.948±0.013 | 0.948±0.007 | 0.913±0.017 |
|  | Hier | 0.945±0.013 | **0.949**±0.010 | **0.917**±0.016 |
|  | IRM | 0.945±0.013 | 0.947±0.009 | 0.909±0.018 |
|  | CLOvE | 0.944±0.008 | 0.949±0.011 | 0.911±0.020 |
|  | VarREx | 0.949±0.013 | 0.948±0.009 | 0.910±0.022 |
|  | VarAUC | **0.950**±0.017 | 0.947±0.008 | 0.912±0.022 |
| Distribution Shift | ERM | 0.731±0.007 | 0.808±0.015 | **0.883**±0.014 |
|  | Hier | 0.727 ± 0.009 | 0.810 ± 0.014 | 0.882 ± 0.020 |
|  | IRM | 0.724±0.017 | 0.806±0.019 | 0.880±0.023 |
|  | CLOvE | 0.745±0.020 | 0.807±0.018 | 0.878±0.020 |
|  | VarREx | 0.729±0.005 | 0.811±0.018 | 0.880±0.026 |
|  | VarAUC | **0.767**±0.008 | **0.838**±0.019 | 0.881 ±0.024 |

where

$$G = \begin{pmatrix} G_{0,0} & G_{0,1} & G_{0,2} \\ G_{1,0} & G_{1,1} & G_{1,2} \\ G_{2,0} & G_{2,1} & G_{2,2} \end{pmatrix}. \tag{63}$$

## C   Comparison to OOD Environment Balancing Methods

Previous methods in the field of OOD generalization (see §2) exhibit several key differences compared to our setting: (i) They address closed-world classification, whereas in zero-shot learning, new classes are encountered. (ii) The presumed shift is typically in the conditional distribution of the data given the class (e.g., the background given the class being a cow or a camel), whereas we consider shifts in the class distribution $P(c)$. (iii) Existing methods often assume that training data comes from various data environments, providing explicit information about how the distribution might shift, while we assume the attribute $A$ causing the shift is unknown.

Despite these differences, in this work we recast class distribution shifts in zero-shot learning into environment balancing OOD setting, by making the following observations. First, when posed as verification methods, zero-shot classifiers in fact perform a binary (closed-world) classification task, predicting whether a pair of data points $x_{ij} := (z_i, z_j)$ belong to the same class $y_{ij} = \mathbb{1}_{c_i = c_j}$.

Note that the distribution of possible pairs $x_{ij} = (z_i, z_j)$ given the label $y_{ij}$ changes with variations in class attribute probabilities, and therefore across synthetic environments $S$. Thus, in this formulation the shift occurs in the conditional distribution of the data given the class $p(x_{ij}|y_{ij})$.

Another distinction lies in data availability: in the setting of closed-world OOD environment balancing methods, a main drawback is the challenge of securing a sufficient number of diverse training environments. This is essential to ensure that a representation performing well on observed environments, will likely perform similarly on unobserved ones. In contrast, our framework allows for the construction of many synthetic environments via sampling.

## D   Additional Empirical Results

### D.1   Additional Simulation Results

**Simulations**   Exact mean and standard deviations matching Figure 4 are provided in table 1.

AUC progress during training iterations and feature importance results for the majority class proportion of $\rho = 0.1$ were shown in the main text. Here, we provide analogous results for $\rho = 0.05$ and $\rho = 0.3$. These are summarized in Figure 8.

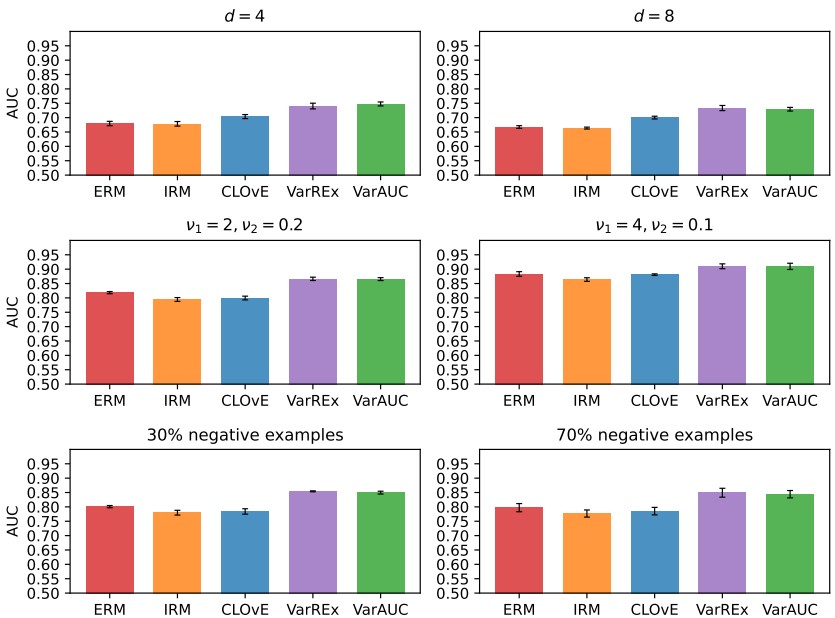

Figure 7: Additional simulation results. Top row: Additional dimensions of the representation. Middle row: additional rations of the attribute variances. Bottom row: unbalanced sets of positive and negative examples. Bars show mean AUC values on the test set across 5 repetitions of the experiment, whiskers show $\pm$ standard deviation.

For $\rho = 0.05$ the convergence results are similar to those obtained for $\rho = 0.1$ – under distribution-shift the two variance based methods show significantly better results compared to other approaches. Our algorithm with the VarREx penalty achieves high AUC values more quickly than the VarAUC penalty, but the VarAUC penalty attains higher accuracy overall. The CLOvE penalty achieves improvement over ERM, but smaller compared to the variance based methods. IRM converges to the same AUC as ERM. In contrast, on in-distribution data all methods perform well.

For $\rho = 0.3$ the distribution shift is milder and therefore ERM performs very well (0.902 AUC is achieved on distribution shift scenario compared to 0.932 on in-distribution setting). Therefore encouragement of similar performance across different data subsets does not benefit the learning process. Slightly better result is achieved with VarREx penalty (0.911).

The analysis of feature importance for $\rho = 0.05$ yields results similar to those for $\rho = 0.1$. At $\rho = 0.3$ the analysis remains mostly unchanged, except that VarREx assigns higher importance to features corresponding to $\nu_0$ (0-5) compared to VarAUC, while in more extreme distribution shifts VarAUC assigns higher importance to the shared features.

### D.2 Additional Representation Sizes, Noise Ratios and Positive Proportions

In §5.1 we explored varying values of $\rho$ in a setting where $\nu^+ = 2$, $\nu^- = 0.1$ ($\frac{\nu^+}{\nu^-} = 20$). We now focus on the case of $\rho = 0.1$ and examine additional representation sizes $p$, and noise ratios ($\frac{\nu^+}{\nu^-} \in \{10, 40\}$). Additionally, we examine the original setting where $p = 16$ and $\nu^+ = 2$, $\nu^- = 0.1$, with varying proportions of positive and negative examples.

The results in Figure 7 show that in all the additional settings our methods provides statistically significant improvement over the baseline. FDR adjusted p-values for multiple comparisons are provided in Table 2.

Table 2: FDR adjusted p-values for the results reported in Figure 7

| Experiment | IRM | CLOvE | VarREx | VarAUC |
|---|---|---|---|---|
| $p = 4$ | 0.7339 | 0.0112 | 0.0003 | 0.0001 |
| $p = 8$ | 0.8552 | 0.0005 | 0.0003 | 0.0001 |
| $\nu_1 = 2, \nu_2 = 0.2$ | 0.9995 | 0.9995 | 0.0002 | <0.0001 |
| $\nu_1 = 4, \nu_2 = 0.1$ | 0.9989 | 0.9971 | 0.0041 | 0.0041 |
| 30% negative | 0.9939 | 0.9939 | <0.0001 | <0.0001 |
| 70% negative | 1.0 | 1.0 | <0.0001 | 0.0002 |

**Experiments**   In Table 3, we provide the means and standard deviations for the experiments detailed in §5.2. Additionally, Table 4 presents the adjusted p-values for assessing the performance increase over the ERM baseline achieved by our algorithm with the explored penalties.

Table 3: Experimental results. Mean and standard deviation of AUC values over 5 repetitions are reported for in distribution scenario ($P_C$), and class distribution shift ($Q_C$). Best result is marked in bold.

| | | IN DISTRIBUTION | DISTRIBUTION SHIFT |
|---|---|---|---|
| CELEBA | **ERM** | $0.826 \pm 0.001$ | $0.666 \pm 0.001$ |
| | **IRM** | $0.843 \pm 0.009$ | $0.659 \pm 0.087$ |
| | **CLOVE** | $\mathbf{0.853} \pm 0.002$ | $0.677 \pm 0.012$ |
| | **VARREX** | $0.834 \pm 0.002$ | $0.676 \pm 0.004$ |
| | **VARAUC** | $0.836 \pm 0.002$ | $\mathbf{0.697} \pm 0.027$ |
| ETHEC | **ERM** | $0.869 \pm 0.004$ | $0.786 \pm 0.030$ |
| | **IRM** | $0.879 \pm 0.004$ | $0.795 \pm 0.034$ |
| | **CLOVE** | $\mathbf{0.888} \pm 0.004$ | $0.800 \pm 0.040$ |
| | **VARREX** | $0.877 \pm 0.007$ | $0.805 \pm 0.033$ |
| | **VARAUC** | $0.872 \pm 0.004$ | $\mathbf{0.838} \pm 0.049$ |

Table 4: Adjusted p-values for one-sided paired t-tests for testing the improvements over the ERM baseline.

| | CELEBA | ETHEC |
|---|---|---|
| **HIERARCHICAL** | 0.0154 | 0.7677 |
| **IRM** | 0.6117 | 0.1290 |
| **CLOVE** | 0.0119 | 0.0383 |
| **VARREX** | $< 0.0001$ | 0.0383 |
| **VARAUC** | 0.0058 | 0.0383 |

### D.3   Analysis of Loss Values

Here we present an analysis of the unpenalized loss after convergence in both real-data experiments. We performed separate analyses on pairs of data points from the dominant type during training (majority), and those from the other type (minority). Additionally, we separated positive pairs ($y = 1$) and negative pairs ($y = 0$). Figure 9 displays histograms illustrating the differences between losses on the training set obtained with the representation learned using ERM ($g_{\mathrm{ERM}}$), and those obtained using our algorithm with VarAUC penalty ($g_{\mathrm{VarAUC}}$):

$$\mathrm{Diff}_{ij} = \ell(x_{ij}, y_{ij}; d_{g_{\mathrm{ERM}}}) - \ell(x_{ij}, y_{ij}; d_{g_{\mathrm{VarAUC}}}).$$

Positive values of the differences correspond to higher losses for ERM.

In both experiments, when examining negative pairs from the minority group, as shown in the top-left histograms, most of the observed differences are positive. This indicates that the ERM losses for these pairs are higher compared to the losses obtained for the representation trained with the VarAUC

penalty. The disparities are smaller for the other three groups: majority negative pairs, minority negative pairs, and minority positive pairs. Among these groups, ERM performs better on positive pairs.

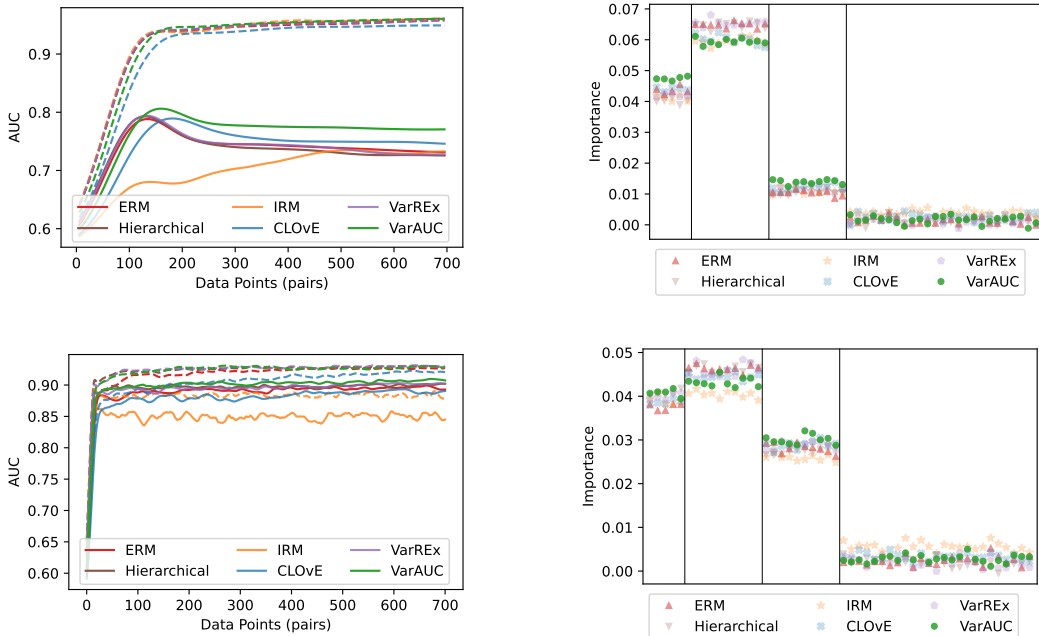

Figure 8: Additional Simulation Results. Top row: $\rho = 0.05$, Bottom row: $\rho = 0.3$. Left: Average AUC progress over 10 repetitions of the simulation. Solid lines correspond to performance on test data (distribution shift scenario), dashed lines show performance on data sampled from the same distribution as training data (in-distribution scenario). Right: Average feature importance results over 10 repetitions.

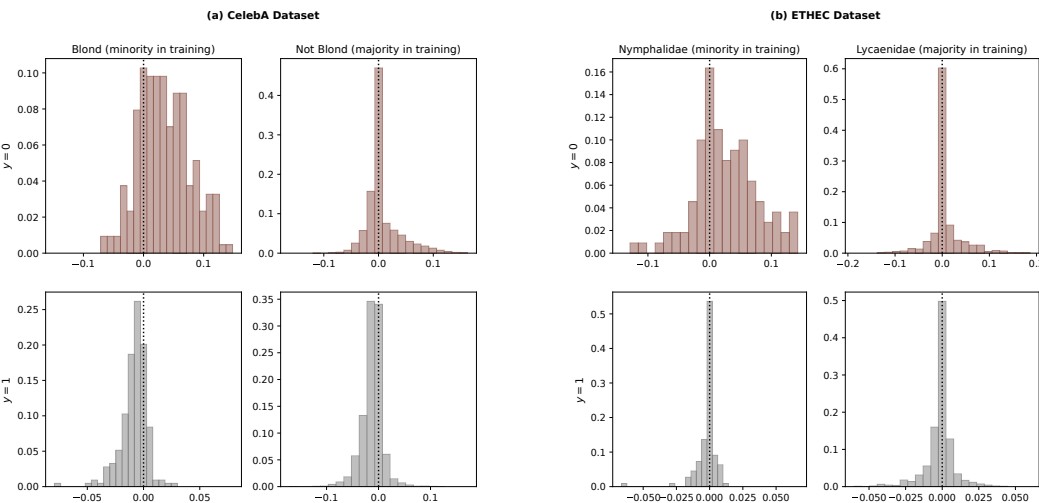

Figure 9: Analysis of Loss Differences. Histograms of differences between ERM and our algorithm with VarAUC penalty are shown for two experiments in separate sub-figures: (a) CelebA dataset, (b) ETHEC dataset. The top rows show differences for negative pairs ($y = 0$), bottom ones show differences for positive pairs ($y = 1$). In each sub-figure the left column corresponds to the minority type and right one to the majority. A dotted black line marks a difference of 0. Positive values correspond to higher losses for ERM.

# E  Datasets

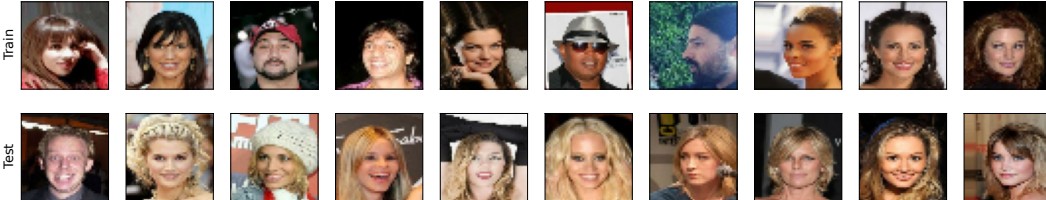

Figure 10: Sample Images from the CelebA Dataset. Top: a random sample of the training data with 95% non-blond people. Bottom: a random sample of the test data with 95% blond people.

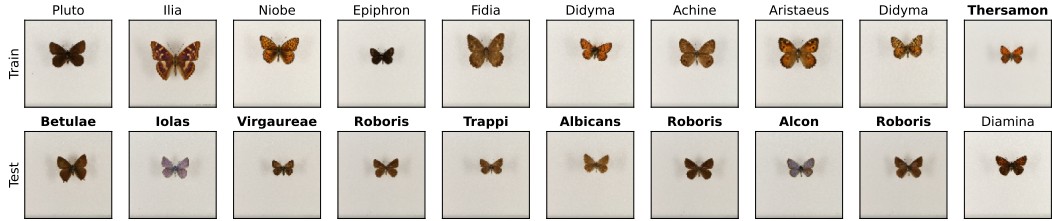

Figure 11: Sample Images from the ETHEC Dataset. Top: a sample of the the training data – 9 species of the Lycaenidae family and 1 from the Nymphalidae family. Bottom: a sample of the test data where the proportion of the families is reversed. Nymphalidae species names are marked in bold.

# F  Implementation Details

A link to a permanent repository with code to reproduce our results is included in the main text.

The data-related parameters of our experiments are described in the main text. In all our experiments we used margin of $m = 0.5$ for the contrastive loss and Adam (Kingma & Ba, 2014) optimizer to train all models.

For the CLOvE penalty we used a Laplacian kernel $k(r, r') = e^{\frac{1}{\text{width}} - |r - r'|}$ with width of 0.4 as originally suggested by Kumar et al. (2018).

For optimization of the VarAUC objective we disregard the finite sample correction $\frac{N_s - n}{N_s - 1}$ in the implementation since $n$ is very small compared to $N_s$. In practice, we minimize the standard deviation instead of the variance in both variance based penalties, and the hyperparameters are reported accordingly.

In our scenario where the attribute of interest is unknown, we generated a synthetic attribute for hyperparameter selection using Principle Components (PC). We ranked examples based on their first PC component values, classifying the top 10% as positive and the rest as negative. Hyperparameters for all methods were chosen via grid-search in a single experiment repetition, ensuring robustness against this synthetic attribute. Notably, the experiments themselves did not involve the PC attribute; instead, they focused on dimension swapping in simulations and attributes like hair color or species family in CelebA and ETHEC experiments.

The grid search produced almost identical hyperparameters for all three $\rho$ values. We observed that performance converged to the same value when employing hyperparameters derived from cross-validation for one $\rho$ value, as those selected for another. Therefore, for simplicity we repeated simulations using the same hyperparameters, determined based on the grid search results for $\rho = 0.1$ (the intermediate parameter value). Similarly, minimal differences in optimal learning rates were observed among the methods within an experiment and therefore a shared learning rate was used for each experiment. To emphasize the improvement of OOD methods over the ERM baseline, we used the learning rate optimized for the ERM method. Large differences were observed in optimal

regularization factors, and therefore these parameters (as well as method-specific parameters) were not shared. All hyper-parameters are reported in Table 5.

All models were initialized with identical weights, and trained on identical data splits.

All the code in this work was implemented in Python 3.10. We used the TensorFlow 2.13 and TensorFlow Addons 0.21 packages. For evaluation we used the `auc` function from scikit-learn 1.2. The CelebA dataset was loaded through TensorFlow Datasets 4.9 and pandas 1.5 was used to process the ETHEC dataset. Statistical tests were performed using `ttest_rel` and `false_discovery_control` functions from scipy.stats 1.11.4. All figures were generated using Matplotlib 3.7.

The IRM implementation was adapted from the source code of the paper, available at `https://github.com/facebookresearch/InvariantRiskMinimization`.

We ran all experiments on a single A100 cloud GPU. For simulations, each full repetition of the experiment (comparing all methods) required on average 2.06 hours. Each repetition on the ETHEC dataset took 7.38 hours on average, and on the CelebA dataset 11.52 hours.

Table 5: Hyper Parameters.

|  |  | ERM | IRM | CLOvE | VarREx | VarAUC |
|---|---|---|---|---|---|---|
| SIMULATIONS | LEARNING RATE $\eta$ | 0.01 | 0.01 | 0.01 | 0.01 | 0.01 |
|  | REGULARIZATION FACTOR $\lambda$ | – | 0.01 | 0.05 | 3.0 | 1.5 |
|  | NETWORK WEIGHT REGULARIZER | – | 0.01 | – | – | – |
| CELEBA | LEARNING RATE $\eta$ | $10^{-5}$ | $10^{-5}$ | $10^{-5}$ | $10^{-5}$ | $10^{-5}$ |
|  | REGULARIZATION FACTOR $\lambda$ | – | 0.1 | 0.085 | 0.01 | 0.2 |
|  | NETWORK WEIGHT REGULARIZER | – | 0.01 | – | – | – |
| ETHEC | LEARNING RATE $\eta$ | $10^{-3}$ | $10^{-3}$ | $10^{-3}$ | $10^{-3}$ | $10^{-3}$ |
|  | REGULARIZATION FACTOR $\lambda$ | – | 0.02 | 0.05 | 0.1 | 0.2 |
|  | NETWORK WEIGHT REGULARIZER | – | 0.01 | – | – | – |

