# OpenReview forum: "Class Distribution Shifts in Zero-Shot Learning: Learning Robust Representations"
_NeurIPS.cc/2024/Conference — NeurIPS 2024 poster_

### Official Review · Reviewer_YoYU · 2024-06-30

**Soundness:** 3
**Presentation:** 3
**Contribution:** 4
**Rating:** 7
**Confidence:** 4

**Summary:**

This paper first investigates the effect of class distribution changes on comparative zero-sample learning by proposing and analysing a class distribution shifts parameter model, leading to the idea that loss minimisation leads to poor performance of representations over the class distribution shifts. Based on this finding, the authors utilise hierarchical sub-sampling and OOD environment balancing methods to obtain robust representations and address the poor performance caused by class distribution changes in zero-sample learning, and experimentally validate the effectiveness of the methods.

**Strengths:**

1- This paper studies the distribution bias problem caused by challenging unknown attributes in zero-shot learning and proposes an effective solution, which is important and innovative for solving the distribution bias problem in zero-shot learning.

2- The structure is clear. It enables the reader to quickly follow the research ideas and understand the content of each section.

3- Figures and tables are clear and accurate. The figures and tables in this paper are concise and clear, effectively support the ideas or conclusions, and enable the reader to grasp the critical information quickly.

4- Comparison and ablation studies are comprehensive. The authors demonstrated the superiority of their method through many experiments and analysed various factors.

**Weaknesses:**

1- Authors should describe their proposed soft-AUC trends in detail and analyse the penalty to help readers understand how they play a role.

2- In Experiment 1, the authors used the attribute blonde hair to shift the class distribution, but we know that some people may be hairless, so can the attribute gender be used to shift the class distribution?

3- The language of this paper needs to be scrutinized and improved. For example, redundant phrases such as "with respect to" (line 32) should be avoided. In addition, there are some grammatical errors that need to be improved, such as "assumes" in line 333 should be changed to "assume", and "leverages" in line 350 should be changed to "average".

4- WRITING DETAILS: Abbreviated nouns should be introduced the first time they appear, such as “OOD”.

**Questions:**

See Weaknesses

**Limitations:**

Yes

---

> ### Author Rebuttal · Authors · 2024-08-06
>
> 1. **Soft-AUC:**
> The use of soft-AUC instead of the standard AUC score is intended to make the complete loss (including the penalty) differentiable, thus enabling gradient-based optimization. As mentioned in lines 218-219, the soft-AUC pointwise converges to the standard AUC score as $\beta$ approaches infinity.
> For a fixed $\beta$, define $f(t_{1},t_{2})=I[t_{1}<t_{2}]$
> and $g_{\beta}(t_{1},t_{2})=\frac{1}{1+e^{-\beta(t_{2}-t_{1})}}$.
> Denoting $t=t_{2}-t_{1}$ we have
> $$\left\Vert f(t_{1},t_{2})-g_{\beta}(t_{1},t_{2})\right\Vert ^{2}=\int_{0}^{\infty}\left(1-\frac{1}{1+e^{-\beta t}}\right)^{2}dt+\intop_{-\infty}^{0}\left(0-\frac{1}{1+e^{-\beta t}}\right)^{2}dt=\frac{1}{\beta}\left[2\log2-1\right].$$
> We will include this calculation in the appendix, along with a graph illustrating the differences, which we have added as Figure 2 in the rebuttal figures file.
> 2. **Gender attribute for the CelebA dataset:**
> In principle, the male/female attribute could be used instead. However, on the CelebA dataset, representations that distinguish well between male individuals also distinguish well between female ones, and vice versa, for a reasonable representation network. Thus, shifts in gender do not call for interventions on the CelebA dataset, as all methods would be equivalent to ERM.
> This is not the case for the attribute of being blond. Our experiments show that the representation learned via ERM on mainly non-blond people (including hairless individuals, as this binary indication distinguishes blond people from all others) fails to separate blond people effectively.
>
> We thank the reviewer for pointing out the typos in 3 and 4.

---

### Official Review · Reviewer_sFfJ · 2024-07-11

**Soundness:** 3
**Presentation:** 3
**Contribution:** 3
**Rating:** 7
**Confidence:** 5

**Summary:**

This paper proposes a robust representation learning method that could assume the shift between seen classes and unseen classes.

**Strengths:**

Good presentation and sound method.

**Weaknesses:**

Lack the experiments on the most popular benchmark of zero-shot learning [1] and comparison to some SOTAs, e.g. [2][3].

[1] Zero-shot learning-the good, the bad and the ugly[C]//Proceedings of the IEEE conference on computer vision and pattern recognition. 2017.
[2] Rebalanced zero-shot learning[J]. IEEE Transactions on Image Processing, 2023.
[3] Transzero: Attribute-guided transformer for zero-shot learning[C]//Proceedings of the AAAI Conference on Artificial Intelligence. 2022.

**Questions:**

See weakness

**Limitations:**

Yes

---

> ### Author Rebuttal · Authors · 2024-08-06
>
> We thank the reviewer for their review and the provided references.
>
> We did not use the datasets mentioned in your references since they either (i) do not have labeled attributes (e.g., the SUN dataset), (ii) the provided attributes correlate with the data-point label such that shifts in them do not affect ERM, making interventions unnecessary (e.g., the CUBS dataset), or (iii) include too few classes (e.g., the AWA2 dataset with only 50 animal classes).
>
>
> Therefore, we performed the real-data experiments on the  CelebA dataset (which is one of the most popular zero-shor benchmarks), and the ETHEC dataset, since both include a large number of classes and labeled attributes in addition to the primary labels (e.g., butterfly family in ETHEC and hair color in CelebA, in addition to species and person identity, respectively).

---

### Official Review · Reviewer_X2Y7 · 2024-07-16

**Soundness:** 3
**Presentation:** 3
**Contribution:** 3
**Rating:** 6
**Confidence:** 2

**Summary:**

Zero-shot learning classifiers face the challenge of distribution shifts, where the distribution of new classes differs significantly from that of the training data. In this paper, the authors introduce a novel algorithm to address this problem by creating robust representations through hierarchical sampling and environment balancing penalization.

Experimental results also demonstrate a performance increase compared to the baseline ERM model on several real-world datasets.

**Strengths:**

1. This paper is well-written and easy to understand.
2. The paper proposes a new model that enables handling unknown attributes for distribution shifts and addresses new classes at test time.
3. The method is tested through both simulations and real-world experiments.

**Weaknesses:**

1. Some parameters need to be clearly defined, for example, $\rho_{tr}$, $\rho_{te}$, and $y_{uv}$ in Eq (4).
2. The proposed method creates multiple environments and computes penalties across them. What is the computational complexity? It's also beneficial to discuss the time complexity of Algorithm 1.
3. Figure 5 and Figure 6 do not straightforwardly show the performance.

**Questions:**

Please see the Weakness section above.

Additional questions:

1. Although the authors mention how to calculate the number of environments, it's better to include an ablation study to test the performance with different numbers of environments.

**Limitations:**

The authors acknowledge the limitations in their paper.

---

> ### Author Rebuttal · Authors · 2024-08-06
>
> We thank the reviewer for their time and comments. Below we address the raised weaknesses and questions:
>
> **Weaknesses:**
> 1. *Definition of parameters:*
> $\rho_{tr}$ and $\rho_{te}$ correspond to the the proportion of type $a_1$ classes in train and test sets correspondingly and are defined in lines 139-140, but we will make their definition more clear. $y_{uv}$ is simply the label for the pair of datapoints indexed by u and v: we need two sets of indices since Eq. 4 treats pairs from the same class ($y_{ij}=1$) and those from different classes ($y_{uv}=0$) separately.
> 2. *Complexity:*
> All OOD approaches from the considered family (see Section 2) involve optimization across multiple environments. Therefore, the complexity of each step includes: (i) the complexity of generating the environments, (ii) the original complexity of a single iteration over the representation neural network multiplied by the number of environments, and (iii) the computation of the aggregated penalty.
> Component (ii) is shared among all approaches from the considered family, while (iii) is negligible compared to training networks, usually involving simple operations like calculating the mean or variance of scalars. Thus, differences between the methods may arise due to (i). However, the additional training time due to the proposed hierarchical sampling, even when compared to training a simple linear representation, is also negligible since hierarchical sampling can be performed offline before training, as shown in the code provided in the supplementary material. We will include a note about this in the manuscript.
> For example, in the species recognition task, standard sampling takes 1.03 seconds, a naive implementation of hierarchical sampling takes 16.6 seconds, and training the representation takes 1 hour and 6 minutes. Thus, the additional time due to hierarchical sampling is less than 0.4% and similar results (all less than 0.5%) are achieved in all our experiments.
> 3. *Exact performances:*
> Simulation performance is shown in Figure 4, with exact performances reported in Table 1 in the appendix. For real-data experiments, exact performances (and corresponding p-values) are provided in Tables 3 and 4 in the appendix.
> Figures 5 and 6 highlight other important aspects: the learned feature importance in the simulations, demonstrating that our method relies on the intended features, and the performance of our method compared to ERM (i.e., the improvement over ERM).
>
> **Question:** Unlike the standard OOD setting, our environments are synthetically generated via sampling, resulting in many more environments (Nc over k) than specified in Equation 10. Therefore, there is no reason to use fewer environments, and using more can only enhance the performance of our method. However, we will include an analysis in the appendix that examines performance as a function of the number of environments used for both ERM and our method.

---

### Official Review · Reviewer_LBt4 · 2024-07-18

**Soundness:** 3
**Presentation:** 3
**Contribution:** 3
**Rating:** 6
**Confidence:** 3

**Summary:**

The paper treats the problem of learning models for zero-shot open-world classification settings (open-world meaning previously unseen classes might appear at test time) that are robust to distribution shifts.

The proposed approach consists of two stages. In the first stage, synthetic environments $S_i$ are sampled from the training data following a hierarchical sampling approach, where first classes and then data pairs according to sampled classes are sampled.
In the second stage, the model is updated to minimise a loss composed of standard ERM and the variance over environment AUC scores.

The benefits of the method are demonstrated on synthetic data, CelebA, and ETHEC (where also on the latter two a distribution shift is introduced synthetically).

**Strengths:**

- The proposed approach to generate synthetic environments through hierarchical sampling seems neat and novel (even though the idea of generating synthetic environments for learning robust models is not novel, see weaknesses)
- Adjusting the performance metric in the variance regularisation term for zero-shot verification, using AUC on embedding distances instead of loss (like in VaRex) seems like a nice way of avoiding performance plateaus and enables better performance in the conducted experiments
- In the experiments, the proposed method shows significant performance gains over ERM
- It is good to see a theoretical derivation of the necessary number of sampled environments to achieve a minimum number of examples from each class in at least one environment during the hierarchical sampling with a certain probability (Section 4.4). I believe this result should be made more prominent in form of a Proposition with a proof (in the appendix).

**Weaknesses:**

- Lack of baselines wrt generation of synthetic environments: The idea of generating synthetic environments to learn models that are robust to distribution shift is not new, and as such the proposed approach should have been compared to existing methods for this. For example, it would be interesting to see how the approach proposed by the authors compares to the approach of the 'Environment Inference for Invariant Learning' paper by Creager et al. (2021).
- The real data experiments are still semi-synthetic in the sense that the distribution shift is introduced synthetically (and is quite stark). I do understand that finding a dataset that has a stark enough distribution shift of one attribute inherently in it is hard or even impossible, and the synthetic shifts are good for highlighting the potential merits of the proposed approach. However, what is missing is a report on the performance of the proposed approach (in comparison to baselines) on the unshifted train and test sets of CelebA and ETHEC, to ensure that there is no performance trade-off.

Minor:
- Figure 2 is never referred to in the text - as a result it is unclear what its purpose is.
- It would be helpful to refer to the result of Section 4.4 already in Section 4.1, when it is claimed that 'hierarchical sampling results in diverse mixtures of any unknown attribute' and that 'smaller subsets with $k < N_c$ classes are likely to exhibit distinct attribute distributions' Section 4.4. backs up these claims.
- L 235 typo at the end of the line -> (10)
- Line 151: $h$ needs to be defined before referring to it in an equation, or immediately after the equation.

**Questions:**

- How was the number of synthetic environments in Experiments 1 and 2 on real data chosen? What values of $\alpha$ from Section 4.4 do they correspond to?

**Limitations:**

Limitations are briefly discussed in the Discussion.

---

> ### Author Rebuttal · Authors · 2024-08-06
>
> We thank the reviewer for their review and questions. Below we address the raised weaknesses and questions.
>
> **Weaknesses:**
>
> 1. *Comparison with Creager et al. (2021):*
> Thank you for referring us to Creager et al. (2021). We found their work very interesting and will cite it in our related work.
> The main difference between our construction of environments and theirs is that Creager et al. (2021) infer the worst-case environments for a fixed classifier (e.g., trained via ERM). In contrast, we consider a shift in an unknown class attribute, which is not necessarily the worst-case and may be uncorrelated or misaligned with the worst-case scenario.
> However, in our synthetic data simulations (but not real-data experiments), we demonstrated our method on the worst-case shift. Therefore, following your suggestion, we compared the performance of our method using our environments versus those of Creager et al. (2021).
> *The results showed no statistically significant improvement over ERM when using Creager et al.'s environments. We included the results in Figure 1 in the rebuttal figures file.*
> Analyzing the results, we discovered that in the context of contrastive learning, Creager et al.'s  assigned environments were almost random, as the optimized soft-assignment q barely changed from the random initialization. We attribute this to Creager et al.'s method being based on the IRM objective, which directly applies to gradients that are known to be noisy in contrastive learning (as discussed at the end of Chapter 5.1). This finding aligns with our broader results, which show that applying the IRM penalty, even on our environments where other penalties provide improvement, does not yield significant improvement over ERM.
>
> 2. *Performance on unshifted distributions:*
> The performance of the proposed approach on the unshifted train and test sets of CelebA and ETHEC you mentioned is indeed included in figure 4 for the simulations and in table 3 in the Appendix and indeed shows no negative effect on unshifted distribution performance. We agree it might be preferable to move it to the main article.
>
> **Question:**
> In both experiments, we set the minimal $\rho$ to 0.15. Therefore, the number of synthetic environments in Experiment 1 (CelebA, with 450 classes after filtering) and Experiment 2 (ETHEC, with 117 classes after filtering) corresponds to $\alpha=0.05$ and $\alpha=0.09$, respectively. We will specify this in the manuscript as well.
>
> **Minor:**
> 1. Figure 2: Thank you for pointing this out, we will add an explanation in the manuscript. Figure 2 shows that the optimal weights of dimensions corresponding to a given type are larger for higher proportions of that type, and that the discrepancy between the weights increases as the ratio of the type variances grows.
> 2. Location of Section Section 4.4: We accept the recommendation to move Section 4.1 immediately after Section 4.4 and thank the reviewer for the suggestion.

---

> > ### Author Response · Authors · 2024-08-07
> > **Clarification for rebuttal comment 1**
> >
> > To clarify this comment regarding Figure 1 in the new figure file:
> > "However, in our synthetic data simulations (but not real-data experiments), we demonstrated our method on the worst-case shift. Therefore, following your suggestion, we compared the performance of our method using our environments versus those of Creager et al. (2021). "
> >
> > Figure 1 shows a comparison of the results of (a) vanilla ERM with (b) the results of training using the VarAUC penalty on synthetic environments that are formed using the method of Creager et al 2021. We can see that using VarAUC on those synthetic environment does not (significantly) improve performance on any of the datasets. This is in contrast to using VarAUC for our hierarchical environments, whose improvement over ERM is shown in the main paper.

---

> > ### Comment · Reviewer_LBt4 · 2024-08-09
> >
> > I appreciate the reviewers answers to my questions and concerns and have updated my score accordingly.

---

### Author Rebuttal · Authors · 2024-08-06

We thank the reviewers for their efforts in reviewing our paper. We address the concerns raised by each reviewer separately. We attach here the rebuttal figures file, which contains the additional figure referenced in our individual responses.

---

### Decision · Program_Chairs · 2024-09-25

**Decision:**

Accept (poster)

**Comment:**

The paper proposes a method for making zero-shot classifiers robust to a particular class of distributions shifts, which the shift is localized in some unknown latent attribute variable. The paper is well-written and the authors have taken a great care in defining the setup precisely. It seems to be a reasonable extension of classical methods in the supervised classification setting that mitigates against label distribution shifts, where each synthetic environment often corresponds to a single class.

The reviewers share an unanimous agreement in favor of accepting the paper, and I agree with that assessment.

I encourage the authors to address all of the comments raised in the reviews.